# Dynamic light filtering over dermal opsin as a sensory feedback system in fish color change

Lorian E. Schweikert [1,2,8] ✉, Laura E. Bagge [3,4], Lydia F. Naughton [5], Jacob R. Bolin[5], Benjamin R. Wheeler[2], Michael S. Grace[6], Heather D. Bracken-Grissom [1,7] & Sönke Johnsen [2]

Dynamic color change has evolved multiple times, with a physiological basis that has been repeatedly linked to dermal photoreception via the study of excised skin preparations. Despite the widespread prevalence of dermal photoreception, both its physiology and its function in regulating color change remain poorly understood. By examining the morphology, physiology, and optics of dermal photoreception in hogfish (*Lachnolaimus maximus*), we describe a cellular mechanism in which chromatophore pigment activity (i.e., dispersion and aggregation) alters the transmitted light striking SWS1 receptors in the skin. When dispersed, chromatophore pigment selectively absorbs the short-wavelength light required to activate the skin's SWS1 opsin, which we localized to a morphologically specialized population of putative dermal photoreceptors. As SWS1 is nested beneath chromatophores and thus subject to light changes from pigment activity, one possible function of dermal photoreception in hogfish is to monitor chromatophores to detect information about color change performance. This framework of sensory feedback provides insight into the significance of dermal photoreception among color-changing animals.

Dynamic color change is a rapid, variable, and context-dependent behavior with shared physiological characteristics among diverse animals[1–10]. Supporting processes such as thermoregulation, sexual selection, and camouflage[6–8], this behavior is employed among animals in habitats as diverse as desert mountaintops and the deep sea[1]. Relative to morphological color change, occurring over days to months[3,4], dynamic or physiological color change can occur within minutes or less[5,8]. These rate differences are based on regulation mechanisms, with the most rapid forms of color change due to neuronal rather than hormonal primary inputs of control[5,8–10]. Animals capable of dynamic color change include cephalopods[11], amphibians[7], reptiles[1], fish[7] and other ectotherms[1], all achieving this feat using

specialized skin cells called chromatophores[1,8,12]. Several major types of chromatophores exist, changing color through the intracellular reorganization of pigment granules, crystals, or reflective platelets[8,12]. For the pigmentary chromatophores of vertebrates, pigment organelles are reversibly aggregated and dispersed within these cells by molecular motors over an extensive microtubule network[5,8]. As a result, incident light strikes either the underlying (typically white) tissue or the exposed pigment (Fig. 1), which gives the skin its light or colored appearance, respectively[8].

Despite diverse evolutionary histories, another commonality among color-changing animals is the intrinsic photosensitivity of their skin and the predicted coupling of this sense to their ability to change

[1]Institute of the Environment, Department of Biological Sciences, Florida International University, North Miami, FL 33181, USA. [2]Biology Department, Duke University, Durham, NC 27708, USA. [3]Torch Technologies, Shalimar, FL 32579, USA. [4]Air Force Research Laboratory/RWTCA, Eglin Air Force Base, FL 32542, USA. [5]Department of Biology and Marine Biology, University of North Carolina Wilmington, Wilmington 28403, USA. [6]College of Engineering and Science, Florida Institute of Technology, Melbourne, FL 32901, USA. [7]Department of Invertebrate Zoology, National Museum of Natural History, Smithsonian Institution, Washington, DC 20560, USA. [8]Present address: Department of Biology and Marine Biology, University of North Carolina Wilmington, Wilmington 28403, USA. ✉e-mail: schweikertl@uncw.edu

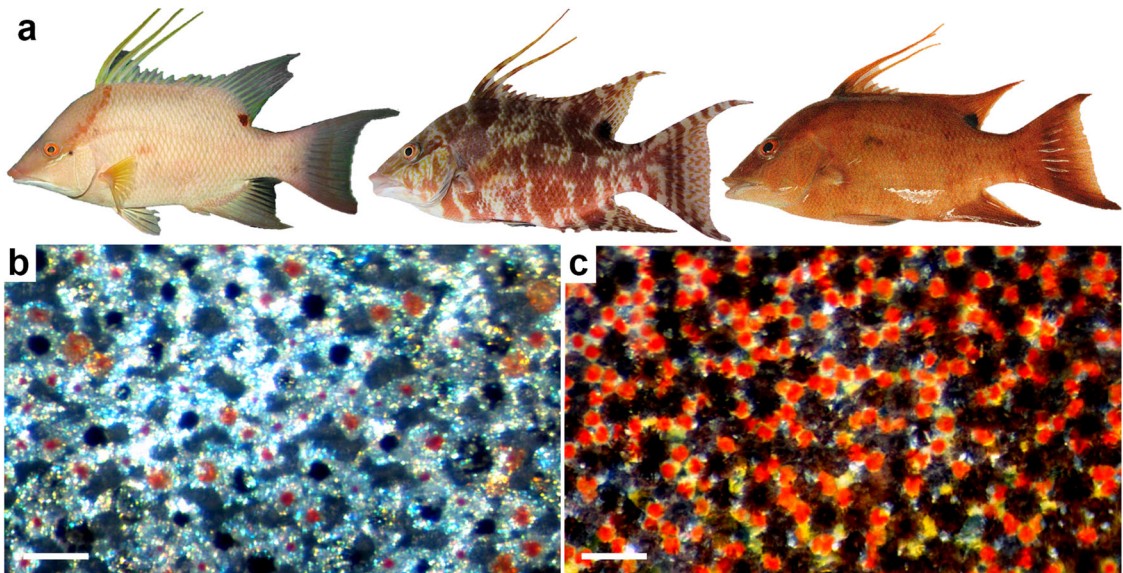

**Fig. 1 | Dynamic color change of hogfish (*Lachnolaimus maximus*).** Hogfish are capable of undergoing rapid changes in skin coloration between at least three chromatic appearances (**a**). Color change is achieved by aggregating (**b**) and dispersing (**c**) pigment granules in chromatophores to generate light and dark cell appearances, respectively. Scale bars equal 100 μm. Panel (**a**) is modified from Schweikert and colleagues (2018)[20].

color[13]. Evidence of dermal photoreception, such as for cephalopods and fish, includes phototransduction proteins (e.g., opsins) of the retina co-occurring in the skin[14–16] and incident light on excised skin patches inducing a color-change response[16,17]. In live animals, however, support is lacking for the direct capacity of dermal photoreception to regulate color change, leaving the function of dermal photoreception in dynamic color change unknown. One hypothesis states that dermal photosensitivity may allow for the regulation of color change independently of inputs from ocular vision[13]. The putative benefits of this strategy include reduced demands of sensory processing for color change or the possibility of light detection outside of the field of view or spectral sensitivity of the eyes[13]. Another hypothesis states that dermal photoreception may locally affect color change within a broader system of control that may coordinate with the central nervous system[15]. This possibility could allow monitoring of chromatophore color change within a feedback system, not unlike the intrinsic photosensitivity of light organs in certain mesopelagic shrimps and the bobtail squid (*Euprymna scolopes*) thought to help regulate outputs of bioluminescence[18,19]. For color change, however, evidence is lacking to support these possibilities, leading to questions about how and why dermal photoreception and color change may be linked.

Our understanding of dermal photoreception in color change is largely based on studies of gene expression (e.g., refs. 20,21), which have indicated that opsins and other phototransduction components expressed in the skin have varying similarities to the phototransduction components of the retina. In color-changing vertebrates, including the hogfish (*Lachnolaimus maximus*), dermal photoreception may incorporate several opsins types and mediate phototransduction via a cAMP-dependent cascade[20,22–24]. Both 'non-visual' type (e.g., melanopsin) and 'visual' type opsins (e.g., RH1, SWS1) have been identified and implicated in chromatophore activation among vertebrates[16,25,26], with a particular role evident for SWS1 (short-wavelength-sensitive-1) opsin. In the hogfish (*Lachnolaimus maximus*)[20], Moorish gecko (*Tarentola mauritanica*)[14], summer flounder (*Paralichthys dentatus*)[26], and Nile tilapia (*Oreochromis niloticus*)[16], regardless of expressing a single or multiple dermal opsins, SWS1 opsin has been consistently identified in chromatophore-containing skin. These studies have shown that, relative to other opsins, SWS1 opsin can have the highest expression levels in the skin (in hogfish and others)[16,20] and that SWS1 activation, at least within in vitro skin preparations of the Nile tilapia,

can directly mediate chromatophore responses to light[27]. Though these studies evidence a relationship between dermal opsins and dynamic color change[13], we still lack knowledge about the functional organization of this system—perhaps critical for understanding the significance of dermal photoreception in living animals.

In addition to studies of gene expression, those examining the arrangement of dermal opsins relative to other components in skin provide key insights into the potential functions of dermal photoreception. We know from a limited number of studies that visual-type opsins can be expressed either within chromatophores or more diffusely, in surrounding cell types[15,16]. In the Nile tilapia (*Oreochromis niloticus*), such opsins have been localized to chromatophores using single-cell reverse transcription polymerase chain reaction (RT-PCR)[16]. Protein localization of opsin and inferred function of dermal photoreception, however, is better described for certain invertebrates. In the inshore squid (*Doryteuthis pealeii*), rhodopsin has been localized to several cell types comprising chromatophore organs: the pigment cells, radial muscle fibers, and sheath cells, which may individually or synergistically respond to incident light[15]. In the ophiuroid, *Ophiocoma wendtii*, other dermal opsins (previously implicated in echinoderm vision)[28,29] were localized to putative photoreceptor cells found between chromatophores, which may serve as screening pigments that confer directionality to dermal photoreception[30]. Despite the widespread prevalence and shared physiological characteristics of dermal photoreception among diverse vertebrates, a similar study examining the optical organization of opsin within skin is lacking for any vertebrate system. Our goal was to conduct such a study, examining design principles of dermal photoreception to better understand the functional significance of this sense in color-changing skin.

The subject of our study, the hogfish (Perciformes: Labridae; Fig. 1), is the largest and most economically valuable wrasse of the western North Atlantic Ocean[31]. Its distinguishing features include hermaphroditic and haremic reproductive strategies[32], which may incorporate color change as a form of social signaling in addition to background-matching camouflage[33]. Post-settlement, both males and females are capable of dynamic color change[33] (within one second or less, Schweikert pers. observation) between at least three chromatic morphs[34]: uniform white, uniform reddish-brown, and a mottled coloration (Fig. 1). Studies are lacking however, on the underlying physiology of hogfish color change.

In this work, we used approaches in immunohistochemistry, confocal and transmission electron microscopy, sequenced-based spectral sensitivity estimation, and microspectrophotometry (MSP) to investigate the physical and optical relationship between SWS1 opsin and chromatophores in hogfish skin. Our results show that SWS1-opsin expression is localized to a morphologically specialized population of cells existing beneath chromatophores and that chromatophore pigment selectively absorbs the wavelengths of SWS1 peak spectral sensitivity. As SWS1 receptors appear subject to light changes from pigment activity (aggregation and dispersion), the predicted function of dermal photoreception in hogfish is to detect these shifts in chromatophore pigment in order to obtain sensory feedback about color change performance.

## Results
### Chromatophores and color change
Three types of chromatophores with differing pigments were identified by light microscopy of *en face* preparations of hogfish skin: black melanophores, red erythrophores, and yellow xanthophores (Fig. 1). Chromatophores were arranged in a horizontal array, existing within a thin dermal tissue layer found on top of the fish's scales. The light white, dark red, and mottled appearances of hogfish skin (Fig. 1a) are achieved by the aggregation and dispersion (Fig. 1b, c) of chromatophore pigment, respectively. Observation of white reflectivity and blue iridescence in the aggregated pigment preparations (Fig. 1b) suggests the presence of leucophores or iridophores in hogfish skin; however, the presence of these chromatophore types has yet to be observed in our analyses by transmission electron microscopy.

### SWS1 immunofluorescence
We performed anti-opsin immunofluorescence to localize SWS1 expression in hogfish skin (Fig. 2). Skin cross sections revealed a dense layer of epidermal cell nuclei overlaying chromatophores and surrounding cell types on top of the fish's scales. SWS1-immunolabeling was localized directly beneath the pigment of chromatophores, not in a continuous layer in skin, but in discrete positions found beneath individual, contiguous chromatophores (Fig. 2a–c). Using differential interference contrast (DIC) microscopy, SWS1 expression was indicated beneath melanophores (Fig. 2c, d); however, the low optical density of erythrophores and xanthophores made it difficult to identify these cells in micrographs by pigment color. Thus, SWS1 expression beneath these chromatophore types was inferred from the adjacent positioning of these cells as shown by light microscopy (Fig. 1) and transmission electron microscopy (Fig. 3). Specificity and sensitivity of the SWS1-opsin antibody were validated by the lack of expression in control preparations and positive labeling of a cone photoreceptor population in cross section of hogfish retina (Fig. 2e, f).

### Skin ultrastructure and SWS1 immunogold labeling
To assess the subcellular ultrastructure supporting SWS1-opsin expression, we conducted transmission electron microscopy (TEM) of ultrathin cross sections of hogfish skin (Fig. 3). Electron micrographs revealed expected elements of fish skin morphology, including the presence of orthogonal collagen lamina found immediately above the chromatophore cells, which showed characteristic differences in the electron density of their pigment[9] (Fig. 3b). Further, micrographs revealed a distinct population of an unknown cell type existing directly beneath chromatophores (Fig. 3). These cells were densely filled with a reticulated membrane, bearing a morphology unlike that known for cell organelles. Section orientation did not change this observation, as the reticulated membrane had the same morphology in both cross-sectional and *en face* planes (Fig. 3 and Supplementary Fig. 1). As was shown for SWS1 immunofluorescence, the membrane-filled cells were found beneath each adjacent chromatophore cell (Fig. 3b and Supplementary Fig. 2).

The two were always coupled, with no instances of membrane-filled cells lacking an overlying chromatophore. Notably, the membrane-filled cells did not appear wider or offset from chromatophores; rather, the margins of both cell types were vertically aligned (Fig. 3b and Supplementary Fig. 2). The SWS1 immunofluorescence (Fig. 2) was, therefore, colocalized to the reticulated membrane structure, suggesting expression of SWS1 opsin within these cells and their function as putative photoreceptors. To further explore this possibility, we conducted anti-SWS1 opsin immunogold labeling of skin cross sections and found positive immunoreactivity in the reticulated membrane structure of these cells (Fig. 4a, b). Though the ultrastructural resolution was limited due to conflict between the tissue treatments that preserve ultrastructure and those that permit antibody binding, positive SWS1 immunoreactivity was observed in the underlying cells but not within control preparations (Fig. 4c, d).

### SWS1 spectral sensitivity estimate
To begin exploring the optical effects of chromatophores overlying the SWS1 receptors, we used a sequence alignment technique to estimate the spectral sensitivity of the hogfish SWS1 opsin. Our goal was to identify the known sequence with the highest homology SWS1 gene to that of hogfish, encoding an SWS1 cone opsin with a previously published wavelength of peak sensitivity ($\lambda_{max}$). Using data from the skin transcriptome reported by Schweikert and colleagues[20], we assessed the similarity of the hogfish SWS1 opsin to archived genes using the BLASTx feature provided by the National Center for Biotechnology Information (NCBI). From this output, the SWS1 opsin with the highest homology sequence to that of hogfish came from the night aulonocara (*Aulonocara hueseri*; family Cichlidae, Genbank accession AY775100.1), which has a known SWS1 $\lambda_{max}$ of 415 nm[35]. The alignment of these two sequences indicated amino acid residues at known SWS1 spectral tuning sites that were nearly identical between the species (Fig. 5)[36]. Of the 13 spectral tuning sites identified[37,38], the only substitutions were S97C and M116V (Fig. 5), which from related studies of site-directed mutagenesis, are predicted to confer small effects on SWS1 spectral sensitivity, shifting $\lambda_{max}$ at most a few nanometers[37,38]. Thus, the hogfish SWS1 opsin has an estimated $\lambda_{max}$ centering on 415 nm, falling within the known $\lambda_{max}$ range for all vertebrate SWS1 opsins (i.e., 360–440 nm)[38].

### Chromatophore microspectrophotometry
The alignment of chromatophores over putative photoreceptors (Fig. 3) indicates that ambient light must first pass through chromatophores before striking SWS1 opsin in hogfish skin, and thus, we were interested in determining the effects of chromatophore pigment on light transmission using microspectrophotometry (MSP). We passed broad-spectrum white light through the dispersed pigment of each chromatophore type ($n = 60$ cells each for melanophores, erythrophores, and xanthophores) to measure transmittance ranging from 400 to 700 nm wavelengths. Light was passed through unpigmented tissue between chromatophores as a reference, allowing the spectral transmittance of the chromatophore alone to be calculated. For all three chromatophore types, transmittance was positively correlated with wavelength. The mean spectra of erythrophores and xanthophores revealed sharp transitions between regions of low and high transmittance, occurring at roughly 550 nm and 488 nm, respectively (Fig. 6). By comparison, melanophores had relatively low transmittance that increased slowly and uniformly. All of the chromatophores types, however, strongly attenuated light over the spectral range of known vertebrate SWS1 opsin sensitivity, and specifically of the predicted hogfish SWS1 sensitivity curve ($\lambda_{max} = 415$ nm) as revealed by visual pigment template fitting, with a ~50%, 85%, and 90% reduction of short-wavelength light transmission shown for xanthophores, erythrophores, and melanophores, respectively (Fig. 6).

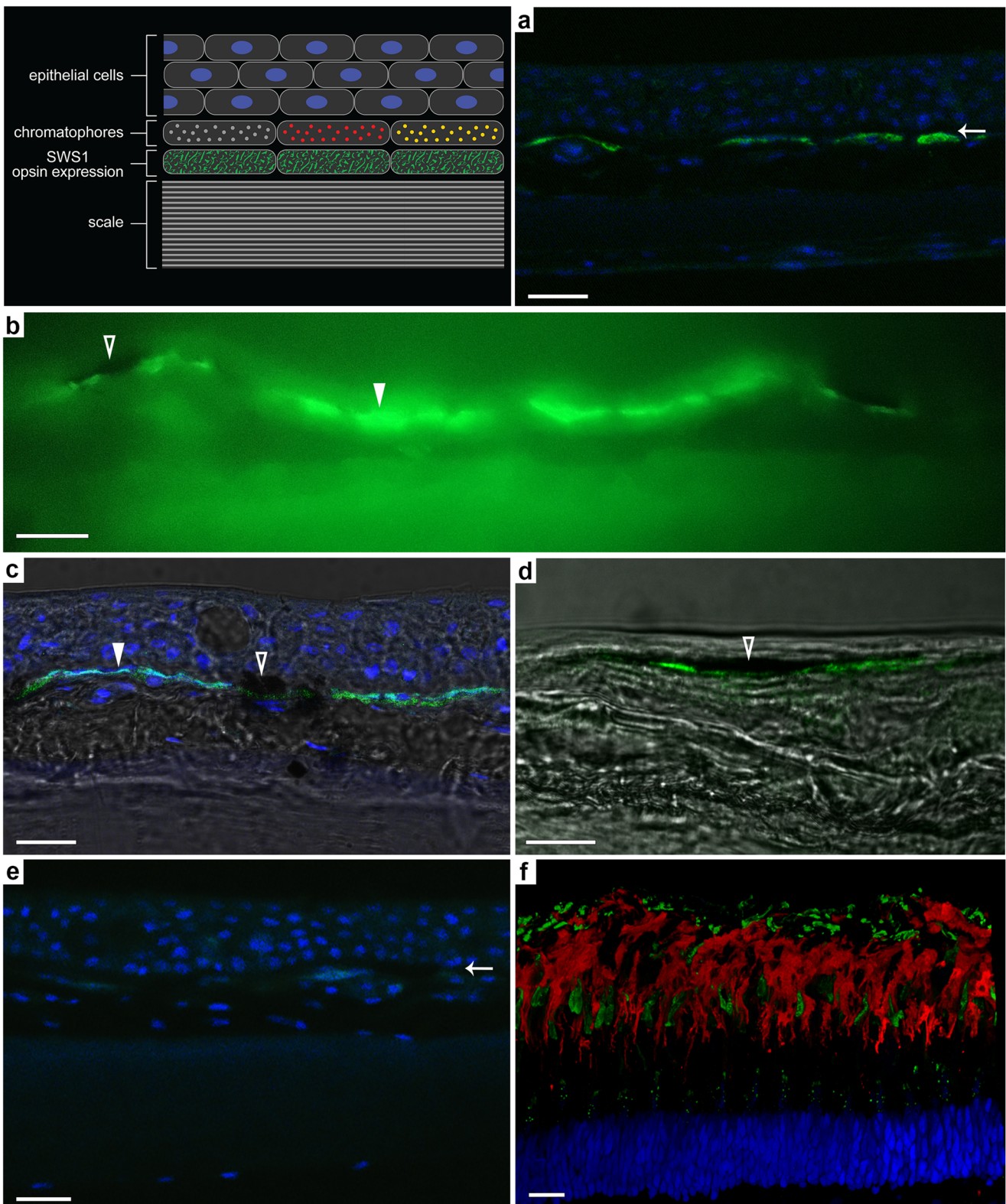

**Fig. 2 | SWS1 opsin expression in hogfish skin and retina.** In skin cross section, immunolabeling of SWS1 opsin (green) is found beneath the chromatophore layer (white arrows; **a**). SWS1 expression is shown beneath melanophores (black-filled triangles) and erythrophores (white triangles; **b**–**d**). In control preparations without primary antibody, immunolabeling of SWS1 opsin in skin cross section is absent (**e**). In retinal cross section, SWS1 immunolabeling (green) of cone photoreceptor outer segments serves as a positive control (**f**). Rod photoreceptor outer segments are indicated by anti-rhodopsin immunolabeling (red), and all cell nuclei are stained with DAPI (blue). All scale bars equal 10 μm.

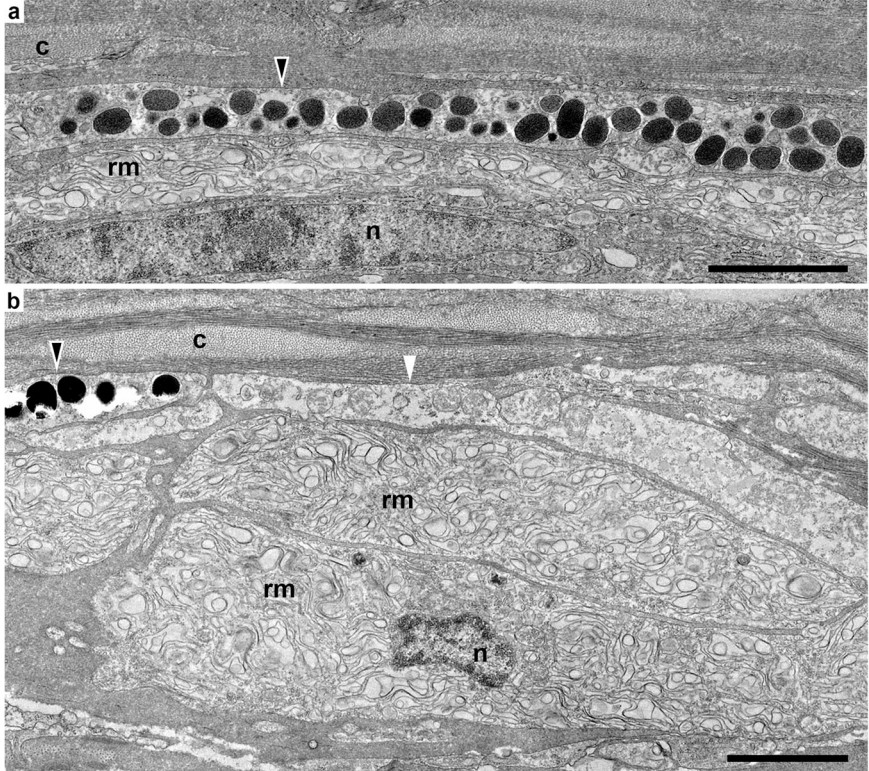

**Fig. 3 | Transmission electron micrographs of skin cross sections in hogfish.** A reticulated membrane structure is contained within a population of cells located beneath chromatophores (**a**, **b**). These membrane-containing cells are shown beneath a melanophore (black-filled triangle) and an erythrophore (white triangle; **a**, **b**). c = collagen fibers, rm = reticulated membrane structure, n = nuclei. Scale bars equal 2 μm.

## Discussion

The expression of SWS1 opsin in hogfish skin suggests dermal spectral sensitivity that coincides with the availability of short-wavelength light that predominates their coral reef habitat[32] but spectrally contrasts with the long-wavelength pigmentation of their skin. This gives way to two possible, though not mutually exclusive, functions of dermal photoreception as previously posited for color change[13,15], which are: (1) to monitor extrinsic information about environmental light, or (2) to monitor intrinsic information about skin coloration. The position of SWS1 receptors beneath chromatophores lends support to the latter possibility, providing insights into how and why dermal photoreception is coupled to color change.

### Functional organization of dermal photoreception

As opsins are transmembrane proteins[39], membrane surface area is correlated with sensitivity. In the vertebrate retina, opsin expression can occur within free-floating discs or laminar invaginations of cell membrane as are found in rod and cone photoreceptors, respectively[40]. Based on the localized SWS1-immunolabeling in hogfish skin (Fig. 2) and previous studies identifying opsins directly within chromatophores (e.g., ref. 16), we expected to see similar modifications of the chromatophore cell membrane. Surprisingly however, we found a distinct and unknown cell type with significant membrane specialization existing beneath the chromatophores, which was both colocalized to and immunogold-labeled as the location of SWS1 expression. Though the undifferentiated morphology of this reticulated membrane is unlike the derived morphology of ciliary photoreceptors, it is not unlike that of cnidarian photoreceptors, for example[41], and may exist to provide a large surface area for opsin expression. The reason for this high surface area (and putative enhancement in sensitivity) is unknown but may relate to maintaining dermal photosensitivity under dim light levels that hogfish may experience when reaching oceanic depths of 20 to 45 m or more[42]. Similar to retinal photoreceptors, these cells may be morphologically and functionally specialized for light reception in the skin. To our knowledge, specialized photoreceptor cells have not been reported in the skin of vertebrates, and thus, the discovery of this cell opens up avenues of research in comparative photoreceptor physiology. Although these data do not rule out the possibility that other non-visual-type dermal opsins (e.g., melanopsin) are expressed within chromatophores or surrounding cells, these findings provide insights into the function of the most abundant opsin in hogfish skin[20].

Our findings are in line with those of Sumner-Rooney and colleagues[30], who found a distinct population of putative photoreceptors between chromatophores in the skin of brittle stars (*Ophiocoma wendtii* and *O. pumila*) that appear subject to light changes from chromatophore pigment migration[30]. In contrast to our study, they found that photoreceptors exist adjacent to chromatophores, with pigment migration creating separate sampling stations over photoreceptors that might confer coarse spatial vision. In hogfish, the putative photoreceptors were found beneath (not adjacent to) chromatophores, with cell boundaries that were vertically aligned (Fig. 3 and Supplementary Fig. 2). The arrangement of this system, with photoreceptors close to one another and directly beneath chromatophores, decreases the likelihood that pigment migration creates separate sampling stations that confer directional vision. This arrangement, however, could allow photoreceptors to detect changes in overlying pigment in order to monitor color change performance—a possibility that is only true if pigment alters the light in a way that is physiologically relevant for SWS1 activation.

Previous site-directed mutagenesis experiments have revealed the key spectral tuning sites that alter SWS1 spectral sensitivity[37,38], and as ciliary opsins are highly conserved, a sequence alignment technique provides a tractable method for $\lambda_{max}$ estimation. Amino acid

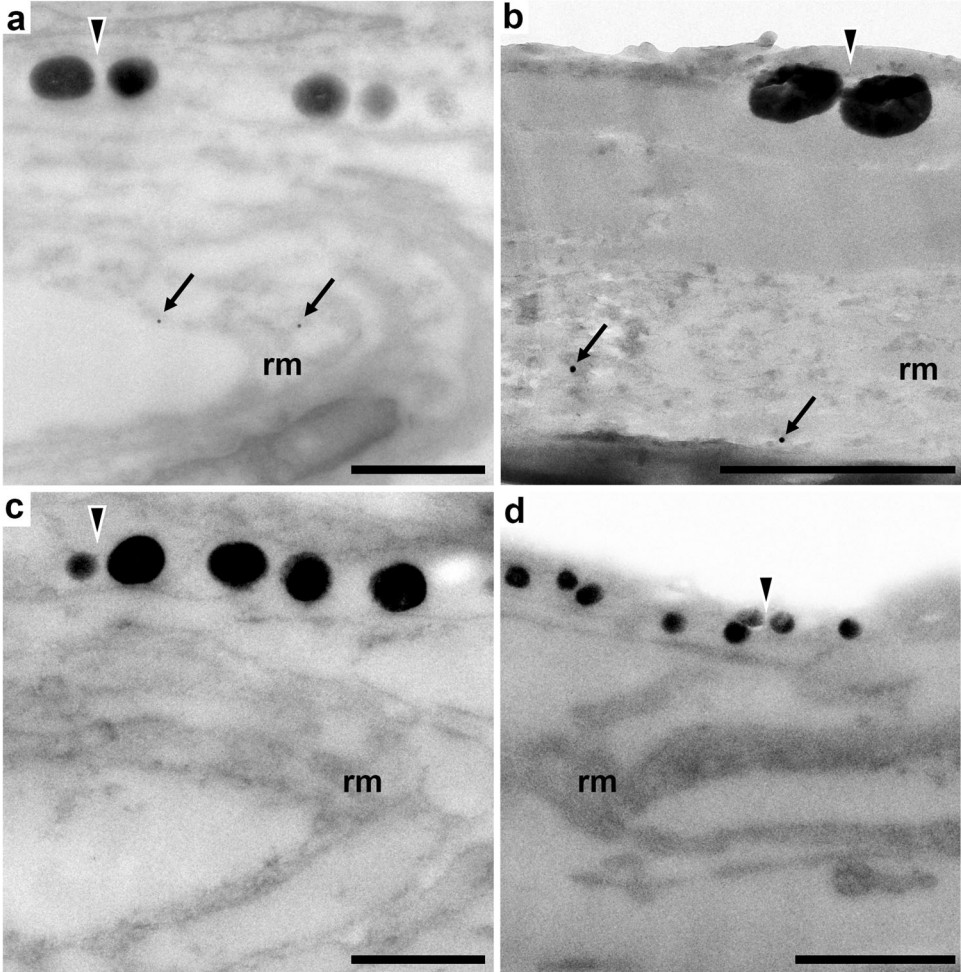

**Fig. 4 | SWS1-immunogold transmission electron micrographs of skin cross sections in hogfish.** Immunogold labeling of SWS1 opsin (black arrows) is shown within the cells beneath melanophores (**a**, **b**). In control preparations without primary antibodies, immunogold labeling of SWS1 opsin is absent (**c**, **d**). Black-filled triangles = melanophores, rm = reticulated membrane structure. Scale bars equal 1 μm in (**a–c**) and 2 μm in (**d**).

substitutions at positions 86, 90, and 93 (relative to a bovine rhodopsin standard) are known to generate the largest changes in SWS1 sensitivity, shifting $\lambda_{max}$ from ultraviolet to blue light[37]. Here, the alignment of the hogfish SWS1 to that of another teleost fish (the night aulonocara) indicated high homology of spectral tuning sites, with only two substitutions at positions 97 and 116. Though the effects of these exact amino acid substitutions have never been independently studied using site-directed mutagenesis, similar switches, at least for position 116, are reported to shift SWS1 $\lambda_{max}$ by 0 to −3 nm[38]. Together, these findings provide a reasonable estimate of the hogfish SWS1 $\lambda_{max}$ at 415 nm, which was fitted to a vitamin A1-based opsin absorbance spectrum that revealed an overall sensitivity range from <350 to 500 nm.

The chromatophore transmission spectra showed that short-wavelength light required to activate the skin's SWS1 opsin is the same light that is selectively absorbed, and thus suppressed, by the pigment of each chromatophore type (Fig. 3). The degree of light attenuation varied between the types according to optical density, with melanophores being the strongest attenuators followed by erythrophores then xanthophores—a finding that is in line with a previous study of chromatophore light transmission in the Japanese Medaka (*Oryzias latipes*)[43]. Their study also showed that each chromatophore type attenuates the ultraviolet portion of the spectrum, following the same trend according to chromatophore optical density[43].

## Functional implications of dermal photoreception

In summary, this system of dermal photoreception in hogfish suggests that dispersion of pigment in chromatophores suppresses short-wavelength irradiation of SWS1 photoreceptors and that aggregation of pigment increases irradiation (and therefore, putative activation) of the SWS1 photoreceptors, making them sensitive to changes in chromatophore color state (Fig. 7). This organization suggests a cellular mechanism for how dermal photoreception governs color change and why it does so, perhaps to provide sensory feedback to chromatophores to fine-tune color change performance. One missing piece, however, is determining how dermal photoreceptors communicate with chromatophores to exert feedback control on skin color change. For example, the activation characteristics of the SWS1 receptors are unknown, along with the synapses and signaling molecules that may connect the two cell types. Though more research is needed, the sensory feedback model offered here helps explain both the lack of behavioral support for the direct control of color change by dermal photoreception and the widespread evolution of this sense across color-changing taxa. Specifically, environmental cues for color change may be captured by the eyes and integrated with feedback information from dermal photoreceptors about skin color state to fine-tune color change output (Supplementary Fig. 3). Such closed-loop feedback systems are common in physiology and behavior[44] and may be required by color change as they are by other outputs where fitness is coupled to the precision of performance[45,46].

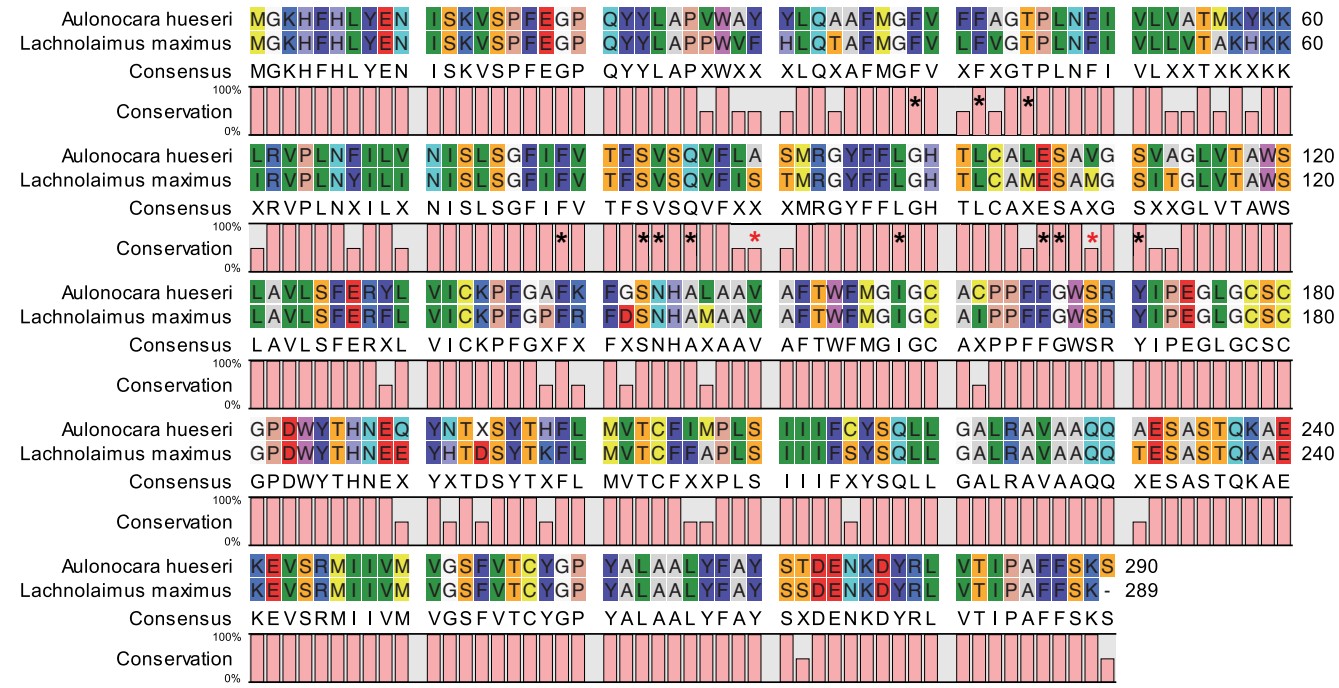

**Fig. 5 | Amino acid sequence alignment of SWS1 opsins between night aulonocara (*Aulonocara hueseri*) and hogfish (*Lachnolaimus maximus*).** The deduced amino acid sequences of the retinal SWS1 opsin from *A. hueseri* (Genbank accession AY775100.1) and dermal SWS1 opsin from *L. maximus* (Genbank Accession: PRJNA386691) are shown. The known spectral tuning sites of vertebrate SWS1 opsins are indicated by the asterisks[37,38], with two amino acid substitutions (red asterisks) identified between the two species, positions S97C and M116V. Percent conservation indicates amino acid similarity.

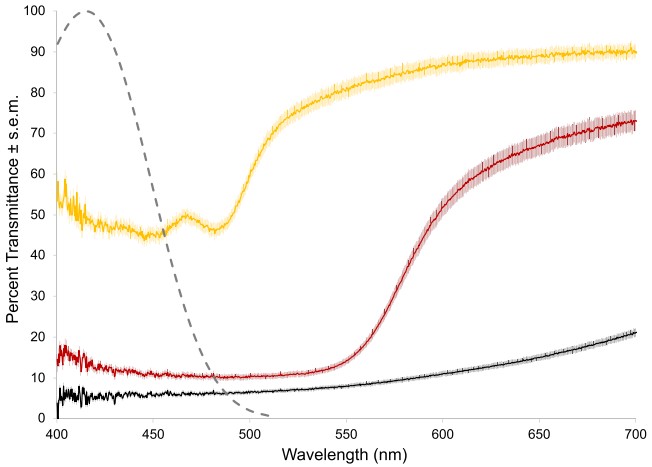

**Fig. 6 | Percent transmittance spectra of the pigmentary chromatophores types in hogfish.** The mean percent transmission (±s.e.m.) of light wavelengths spanning 400–700 nm is shown for the dispersed pigment of melanophores (black line), erythrophores (red line), and xanthophores (yellow line); $n = 60$ per cell type. A fitted template for a vitamin-A-based photoreceptor action spectrum, with a peak wavelength of sensitivity ($\lambda_{max}$) at 415 nm, is indicated by the gray dashed line.

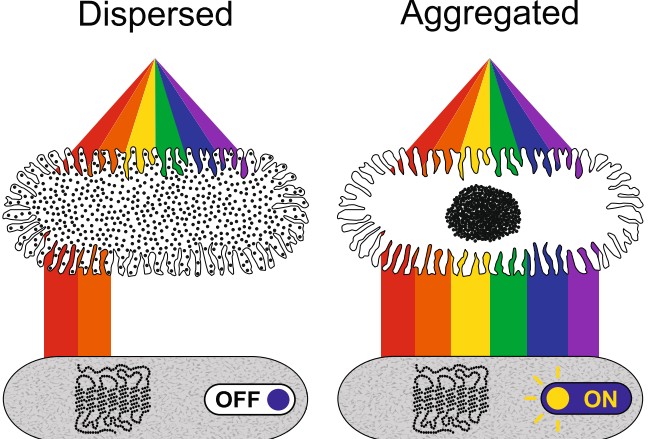

**Fig. 7 | A conceptual diagram of the functional relationship between chromatophores and putative photoreceptors in hogfish skin.** Dispersed chromatophore pigment suppresses short-wavelength irradiation of SWS1 receptors (left), whereas aggregated pigment permits short-wavelength irradiation (and, therefore, putative opsin activation) of SWS1 receptors (right). The predicted functional significance of dermal photoreception is, therefore, to monitor shifts in chromatophore pigment in order to detect feedback information about color change performance. Illustrated by M.D. Smith. Chromatophores = white cells at the top; SWS1 dermal photoreceptors = gray cells at the bottom.

Lastly, the physiological characteristics of dermal photoreceptors, such as their cellular activation characteristics and vitamin-A chromophore content for visual pigment function[40], remain unknown. We can look to retinal photoreceptor physiology, however, to gain some insight into how this system might work. For example, rod and cone characteristics, such as cell stimulation over graded membrane potentials that scale with exposure to light[47], would be particularly relevant to this system where the intensity of incident light upon dermal photoreceptors is dependent on the degree of pigment aggregation and chromatophore pigment type. Again, though dermal photoreception in hogfish and other species requires further research, our findings suggest a future area of study for extraocular photoreception related to sensory feedback, providing a framework for understanding the widespread prevalence of dermal photoreception in the skin of color-changing animals.

## Methods

### Animals

The study specimens were hogfish (*Lachnolaimus maximus*; family Labridae) ranging in total length from 16.5 to 35.5 cm ($N = 16$ fish, total). Hogfish is a protogynous hermaphroditic reef fish, switching from female to male as required at roughly 30.5 cm fork length[32]. Thus, the specimens included in this study are primarily female, representing subadult to adult life-history stages. Wild-caught hogfish were collected under a Florida Fish and Wildlife Conservation Commission special activity license (SAL-16-1822A-SR), by the approval of the Institutional Animal Care and Use Committees at Duke University (protocol #A233-16-10), Florida International University (protocol #IACUC-19-024), and the University of North Carolina Wilmington (protocol # A2020-016). Commercially-obtained hogfish were purchased from Dynasty Marine Associates (Marathon, FL) and Gulf Specimen Marine Laboratories, Inc. (Panacea, FL). All animals were humanely euthanized by either overdose of MS-222 (Tricaine) or eugenol (clove oil) according to approved IACUC procedures. For microspectrophotometry only, fresh carcasses provided tissues of adequate quality, which were obtained from recreational fisherman via Wrightsville Beach Diving Spearfishing Charter (Wrightsville Beach, NC).

### Light microscopy

We used light microscopy to identify the types of pigmented chromatophores present in hogfish skin ($n = 3$ fish). Whole-mounted scales were placed under an Olympus SZX16 Stereomicroscope and imaged using Olympus DP71 camera (Olympus Scientific Solutions Americas, Waltham, MA). Images were taken of hogfish scales that were light and dark in appearance. Chromatophores were identified by morphology and pigment color.

### Anti-opsin immunofluorescence

The scales of hogfish are covered with a thin layer of integument that contains the chromatophores cells that mediate color change. Thus, to examine SWS1 opsin expression in hogfish skin ($n = 5$ fish), scales were selected at random from different body regions (e.g., dorsal, ventral, and caudal body regions) and processed using conventional immunohistochemical techniques. To serve as a positive control, the left eye of a hogfish ($n = 1$) was taken and processed with the cornea, lens, and humors removed. These samples (either scales or eyecup) were then fixed in a solution of 4% paraformaldehyde in 1X phosphate-buffered saline (PBS), pH 7.4. Following a minimum of 48 h of fixation, samples were transferred to 25% sucrose in 1X PBS for cryoprotection, then embedded in Tissue-Tek O.C.T. compound at a 20 °C. Frozen cross sections (18-μm thick) were cut on a Leica CryoCut 1800 cryostat, thaw-mounted onto gelatin-coated glass microscope slides, and dried at room temperature overnight. Slides were then placed into fixative for 1 h, followed by four 15-min washes in 1X PBS (pH 7.4). The primary antisera (details below) were diluted in PBS containing 0.25% λ-carrageenan, 1% bovine serum albumin, and 0.3% Triton X-100 and applied to the slides for overnight incubation (minimum 8 h) at room temperature. After four 15-min rinses in PBS, slides were incubated for 1 h at room temperature with a fluorophore-conjugated secondary antiserum. Following four rinses in PBS, slides were coverslipped with Slow-fade Gold mounting medium with DAPI nucleic acid stain (Life Technologies, Grand Island, NY) and imaged either on a Nikon C1Si upright laser-scanning confocal microscope (Nikon Instruments, Melville, NY) or on a Leica SP8 upright laser-scanning confocal microscope (Leica Microsystems, Buffalo Grove, IL). Preparations also were imaged on a Zeiss Axioskop2 light and epifluorescence microscope mounted with an 89 North PhotoFluor LM-75 fluorescence light source (Carl Zeiss Microscopy, Peabody, MA) and a Pixera Penguin 600CL camera (Pixera Corporation, Santa Clara, CA). Images were post-processed in Adobe Photoshop to enhance contrast to intact images and add annotations.

All primary and secondary antisera were commercially obtained. The SWS1 opsin antiserum was raised against a recombinant human SWS1 immunogen (1:200-1000 concentration, polyclonal, EMD Millipore catalog# AB5407) and has known specificity and cross-reactivity to SWS1 opsins in diverse species (e.g., refs. [48,49]). For the retina, rod opsin (rhodopsin; RH1) antiserum (1:500 concentration, monoclonal, EMD Millipore catalog# MAB5316) was used to counterstain rod photoreceptor outer segments. The primary antisera were labeled with secondary antisera that were conjugated to Alexa Fluor fluorescent dyes (1:500, Thermofisher Scientific catalog # A-11008 and A-21422).

### Transmission electron microscopy

Scales taken from hogfish ($n = 2$ fish) originally fixed as described above were then immersed in modified Karnovsky's fixative (2.5% glutaraldehyde and 2% paraformaldehyde in 0.15 M sodium cacodylate buffer, pH 7.4) for at least 4 h, postfixed in 1% osmium tetroxide in 0.15 M cacodylate buffer for 1 to 2 h and stained en bloc in 2% uranyl acetate for 1 h. Samples were taken through two iterations of serial dehydrations in ethanol (50%, 70%, 90%, 100%), embedded in Durcupan epoxy resin (Sigma-Aldrich, St. Louis, MO), sectioned (both transverse and sagittal) at 50 to 60 nm on a Leica UCT ultramicrotome, and picked up using Formvar and carbon-coated copper grids. Sections were stained with 2% uranyl acetate for 5 min and Sato's lead stain for 1 min. Grids were viewed using a Tecnai G2 Spirit BioTWIN transmission electron microscope equipped with an Eagle 4k HS digital camera (FEI, Hillsboro, OR).

### Anti-opsin immunogold labeling

Hogfish scales ($n = 2$) were fixed in 4% paraformaldehyde in 1x PBS for 24 h and stored in 1x PBS at 4 °C before processing. Samples were taken through a dehydration series with ethanol (50% and 70% for 15 min, 80% for 10 min), infiltrated with a 2:1 mixture of LR White Resin (Electron Microscopy Sciences, Hatfield, PA) to 80% ethanol, and embedded in resin with four changes of LR White (1 h, overnight, and twice for 30 min). Resin was cured in Beem capsules in a vacuum oven at 50 °C for 4 days. Cross sections of the embedded scales were cut at 90 nm using a Lecia UC7 ultramicrotome and picked up on Formvar-coated nickel mesh grids. Grids were floated on drops of the SWS1 primary antiserum described above (EMD Millipore catalog# AB5407) diluted in 1% bovine serum albumin (BSA) in 1x PBS (1:300) for 2 h at room temperature in a humidity chamber. After washing on drops of 1x PBS four times for 5 min each, grids were floated on drops of a 25-nm colloidal gold secondary antiserum diluted in the BSA solution (1:40, Electron Microscopy Sciences, Hatfield, PA) for 2 h at room temperature in a humidity chamber. As expected, the use of these relatively large gold particles permitted the low-magnification imaging required to visualize the target cells but resulted in sparse labeling by the secondary antibody as explained by Cornford and colleagues (2003)[50]. Grids were again washed in 1x PBS and subsequently in DI water. Sections were imaged using a Tecnai G2 Spirit Bio Twin transmission electron microscope at 80 keV with an Eagle 2k HR 200 kV CCD camera (FEI, Hillsboro, OR). Images were post-processed in Adobe Photoshop to enhance contrast and add annotations.

### Spectral sensitivity estimation

We evaluated the deduced amino acid sequence of the SWS1 opsin found in hogfish skin to estimate its wavelength of peak sensitivity ($\lambda_{max}$). Previous studies using site-directed mutagenesis and other methods have identified the amino acid sites that affect spectral tuning of ciliary opsins (such as SWS1)[37,38]. Evaluating amino acid substitutions at these positions relative to their known consequences on opsin sensitivity allows $\lambda_{max}$ values to be inferred. Using data from the hogfish skin transcriptome reported by Schweikert and colleagues[20], we first assessed the SWS1 opsin gene using the BLASTx feature provided by the National Center for Biotechnology Information (NCBI).

From this output, the animal with the highest homology SWS1 gene to that of hogfish, which also had a known SWS1 $\lambda_{max}$, was identified as the night aulonocara (*Aulonocara hueseri*; family Cichlidae, Genbank accession AY775100.1). The spectral tuning sites of ciliary opsins are commonly reported relative to the positions of a bovine rhodopsin gene standard. To locate these tuning sites, the deduced amino acid sequences of SWS1 from hogfish and the night aulonocara were aligned to bovine rhodopsin using CLC Viewer Software (Qiagen, Redwood City, CA). We then compared the amino acid residues at all known SWS1 spectral tuning sites between these fishes to estimate the $\lambda_{max}$ of SWS1 in hogfish skin. The full absorbance spectrum was calculated from the $\lambda_{max}$ using the template found in Stavenga and colleagues (1993)[51].

### Microspectrophotometry

We measured the transmission spectra of hogfish chromatophores types (i.e., melanophores, erythrophores, and xanthophores) using microspectrophotometry (MSP). Scales were selected at random from different body regions (e.g., dorsal, ventral, and caudal body regions) for analysis within 12 h of hogfish euthanasia ($n = 3$ fish). Skin tissue was removed from scales using a razor blade and was then mounted on a #1.5 glass coverslip preparation (Electron Microscopy Sciences, Hatfield, PA) using 30% glycerol in 0.1 M sodium phosphate buffer, pH 7.4. A ring of silicon grease was placed around the tissue, then a second coverslip was pressed on top of the preparation. MSP was performed on a Nikon Diaphot-TMD inverted compound microscope (Melville, NY). A 20-Watt quartz tungsten halogen lamp (Optometrics LLC, San Francisco, CA) provided white light, which was passed through a 400-μm diameter fiber (Ocean Optics, Dunedin, FL, USA) and focused by a condensing objective through a single chromatophore cell (to the margins of dispersed pigment). The transmitted light was collected by a Zeiss 16x Neofluar microscope objective before passing a 1-mm diameter fiber (Ocean Optics Inc., Dunedin, FL) connected to a USB2000 spectrometer (Ocean Optics Inc., Dunedin, FL). Reference scans were taken in areas of unpigmented tissue, in the space between chromatophores with aggregated pigment. Transmittance spectra of each chromatophore cell type (20 cells per type and fish and thus, $n = 60$ cells per chromatophore type) were measured for each fish using OceanView Software (v1.6.7; Ocean Optics Inc., Dunedin, FL). Spectra for each cell type were averaged across fish at ~0.3 nm optical resolution to generate transmission spectra of hogfish chromatophores spanning from 400 to 700 nm. The spectra were averaged using Microsoft Excel and were visually inspected in order to describe transitions between regions of low and high transmittance. The reported wavelengths reflect the point where the slope of these transitions would intersect with the x-axis.

### Statistics and reproducibility

Experiments associated with each method were replicated multiple times, independently. For light microscopy, micrographs were collected across six independent sample preparations across three fish. For immunofluorescence, experiments were run three times independently on separate days at the Florida Institute of Technology and replicated an additional two times at Duke University. The TEM and immunogold images were collected from two separate sample preparations, which had been analyzed at UCSD and UNCW, respectively. For microspectrophotometry, data were collected across three fish that had been sampled over separate days. Spectra for $n = 60$ cells/chromatophore type were collected using no less than 10 scales sampled from each fish body.

### Reporting summary

Further information on research design is available in the Nature Portfolio Reporting Summary linked to this article.

## Data availability

As part of this study, we examined the SWS1 opsin gene sequence of *Aulonocara hueseri* (Genbank accession AY775100.1) and aligned it to the SWS1 opsin sequence of *Lachnolaimus maximus* (Genbank Accession: PRJNA386691). All other data included in this study are available in the main text and supplemental materials.

## Code availability

There is no code to report.

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

## Acknowledgements

We thank the microscopy facilities of Duke University, Florida Institute of Technology (FIT), University of North Carolina Wilmington (UNCW) and the University of California San Diego (UCSD) Cell and Molecular Medicine (CMM) Department for shared resources and staff support. At UCSD/CMM, we thank M. Farquhar for the use of the TEM facility, Y. Jones for help with sample preparation, and T. Meerlo for general assistance. We also thank L. Elliot and A. Taylor for research support in the Richard M. Dillaman Bioimaging Facility at UNCW. We also thank H.F. Nijhout (Duke) and T. Frank (Nova Southeastern University) for loaning equipment for this study. We acknowledge M. Bolton, T. Holford, N. Kamasawa at the MPFI for Neuroscience and J. Fasick (University of Tampa) for providing helpful insights into our study. We thank Captain C. Slog for providing specimens, D. Kimberly for hogfish photographs, and M.A. Schweikert for help in hogfish collection. We thank W.M. Kier, E.M. Caves, R.R. Fitak, A.L. Davis, and D.E. Speiser for comments on earlier versions of the manuscript. We also thank the Center for Marine Sciences at UNCW, namely R. Moore and J. White for help with animal husbandry and M.D. Smith for scientific illustration. This is contribution #1595 from the Institute of Environment at Florida International University. This study was supported by Duke University's Charles W. Hargitt Fellowship, the FIU CASE Distinguished Scholar Award, and monies from the FIU Center for Coastal Oceans Research in the Institute for Environment awarded to L.E.S. Additional support came from the NSF Division of Environmental Biology Grant # 1556059 awarded to H.B.G. This study was also supported by the Marine Biological Laboratory Neurobiology Post Course Research award to L.F.N. Further, this material is based on research sponsored by AFRL/RW under agreement number FA8651-22-0036 awarded to L.E.S. The U.S. Government is authorized to reproduce and distribute reprints for Governmental purposes

notwithstanding any copyright notation thereon. The views and conclusions contained herein are those of the authors and should not be interpreted as necessarily representing the official policies or endorsements, either expressed or implied, of AFRL/RW or the U.S. Government. This work is approved for public release (Distribution A), case #AFRL-2023-2347.

## Author contributions

L.E.S. and S.J. conceived of the study and experimental design; L.E.S., L.F.N. and J.R.B. immunofluorescence (aided by B.R.W., M.S.G., and H.B.G.); L.E.S. carried out the sequence analysis and micro-spectrophotometry (aided by S.J.); L.E.B. and L.F.N. completed the transmission electron microscopy; B.R.W. managed hogfish procurement and husbandry; L.E.S. drafted the manuscript, with L.E.S. and L.F.N. conceptualizing Fig. 7; All authors contributed to the interpretation of the results and completion of the final manuscript.

## Competing interests

The authors declare no competing interests.
