## [Peer Review File · Nature Communications]

Dynamic Light Filtering over Dermal Opsin as a Sensory Feedback System in Fish Color ChangeReviewers' Comments:

Reviewer #1:

Remarks to the Author:

This manuscript investigates dermal photoreception in the hogfish, characterising for the first time in an vertebrate the optical organisation of dermal opsin. The authors demonstrate that dermal photoreceptors are positioned underneath chromatophores, suggesting that the photoreceptors may detect colour change performance. The paper is very well written and is a thorough investigation of this system. I really enjoyed reading the paper and I think it is well suited to Nature Communications (including the format – results are easily understood with the methods at the end).

My comments/suggestions are all minor. Mostly, I would like a little more information about the ecology and broader function to better understand how this system may work.

1) Some more information about hogfish would be useful, such as if they use colour change for communication, camouflage or both?

2) How do animals use dermal photoreceptors? The start of the discussion really clearly lays out potential functions of dermal photoreceptors, which I think could be either moved to the introduction or stated more clearly in the introduction. Following from this, how do animal integrate the information from dermal photoreceptors? For example, if their colour change is for camouflage, is the idea that the animal integrates information from the eye about background colour with information from the dermal photoreceptors about its own colour to improve background match? Perhaps this is obvious, but it wasn't entirely clear to me.

Additional Comments:

Line 21: Would colour change due to changes in nanostructures also be considered dynamic colour change? Specifically, I'm thinking of several Charidotella leaf beetles that can change colour by fluid movement between layers in the cuticle that produce structural colour. As far as I know, chromatophores aren't involved.

Line 33-35: This sentence could be written a little more clearly for people less familiar with the content.

Line 62: Do you have space to spell out the function a little more?

L160: This is unclear whether you are referring to the current research or previous research. It is clear in the introduction that it has been discovered before, but could be a little clearer here since it's the opening sentence.

Line 213: Just out of interest, how many nanometers (roughly) are these large shifts? I am wondering if what the implication would be if one of the sites does happen to produce a large shift. I am guessing the outcome will be the same – the absorbance will still be within the region of low transmittance for the chromatophores.

Line 388: Methods are well written! Only comment is, is there a reason why this value (lambda cut-off) was estimated rather than calculated?

Figure 7: Love this, really clear!

Reviewer #2:

Remarks to the Author:

Review Nature Communications, Schweikert et al Dynamic light filtering over dermal opsin in fish

This is a very interesting paper that reports the potential cellular mechanism by which fish chromatophore pigment activity and resultant color change could be monitored by dermal photoreception. This could provide sensory feedback to fine-tune color change for a variety of behavioral functions.

Comments for the authors:

Abstract: first sentence. Not sure whether referencing is allowed in Nature Comm, but the first sentence would benefit from a reference, since this forms the basis for this paper.

Introduction: First two sentences use the same two references. The field is much broader than represented by those references, and it would be valuable to include relevant literature on color change in fish in particular, paying attention to the overall mechanisms of color change. Pigment changes are not the only changes that occur in fish, and differentiation between pigment and structural color changes are important to point out. This is especially relevant to line 22, that suggests color change is achieved by chromatophores.

Reference 3 is a very old book that is of course useful but the authors may want to consider adding more recent references that deal specifically with fish chromatophores (there are other relevant papers by Sköld et al, for example, but also others).

A short section on the color change in this fish species would be good. What are color changes used for? How fast are they? What are the behavioral patterns of this fish species?

Results: How fast are the color changes? Are reflector cells involved? Fig. 1b shows some blue iridescent looking structures that seem to suggest that structural coloration is involved? I would like to see the spectral data (Fig. 6) in the first paragraph of the results, incorporated into Fig. 1, possibly.

The immunogold labeling images are not terribly convincing – perhaps you could comment on this in the paper, if you do not have more convincing images? Maybe show several images that prove labeling?

Line 108 – the cells were found beneath each adjacent chromatophore cell: are these cells absent from places that are lacking chromatophores? Such places may be hard to find in this fish, which I'm not completely familiar with, but perhaps an area between the fin spines, ventral belly, underneath throat?

MSP data: I would move this up to the beginning of the results section, and incorporate the Fig. 6 into Fig. 1. Could you add how many pigment granules were measured in the transmission graphs? Presumably, n=60 refers to the number of chromatophores? It may be worth adding images to the figure that show the pigment coverage that was measured?

Discussion: Lines 161-163 talks about the short-wave sensitivity of the photoreception and attempts to bring this into ecological context. A short-wave sensitivity at 415 nm is, in my opinion, of very limited use. All of the chromatophores are long-wavelength pigments and it seems that long-wave photosensitivity would make more sense. Perhaps you could expand on this? Unless, of course, the short-wave sensitivity is involved with other structures, e.g., short-wave structure colors (see Fig. 1b). Also, I found the low transmission of the chromatophore pigments very intriguing; approx. 50% for yellow xanthophores, 5-15% for melanophores and erythrophores. Under what conditions (if so) do they disperse separately? It seems that they disperse together (Fig. 1b), or are these fish spp. able to differentially disperse different chromatophore classes? One possible scenario could be that the differential activation of the photosensitive cells provided by the differential transmission from the chromatophores could be the trigger for fine tuning of color change? See also your discussion points in lines 202-208.

Lines 222-225, see my last comment. How could this help with regulating color change? It's worth

expanding here, in my opinion.

Lines 236-237. Fine-tune color change performance. This sentence (while it can stay where it is), needs to be explained somewhere in the introduction. At this point, the reader does not fully understand why this fish species changes color, and why the color may need to be fine tuned. While some color changes in fish are fast, they usually take hours or days to complete. It would be good to expand on this in the context of skin photoreception.

Reviewer #3:

Remarks to the Author:

Key results:

Schweikert et al. address the question of how a dermal photoreception system may interact with chromatophores to help explain how hogfish color-change is controlled. They found SWS1 opsin expression in a layer of cells that lie beneath the chromatophores, rather than within the chromatophores themselves using a combination of immunohistochemistry and TEM. The authors estimated that the SWS1 opsin in hogfish skin has a peak sensitivity at short wavelength, 415nm light, as the sequence is highly similar to SWS1 opsins from the retinas of other fish species. They found that all 3 types of chromatophores in hogfish skin interfere with the amount of short wavelength light that would otherwise reach the photoreceptors when the chromatophore pigment is dispersed, although the extent of this interference differs between chromatophores of different colors.

Validity:

For the most part, the interpretation of the data and conclusions seem sound. That said, given my understanding of opsin phototransduction cascades and their physiological effects within photoreceptor cells, I believe the model presented in the paper for how the chromatophores interact with photoreceptors is not accurate. Typically, when ciliary opsins are exposed to light, they cause the cell to hyperpolarize, effectively turning it "off", and the cells depolarize (turn "on") in the dark. Based on this understanding, the model presented in the discussion and illustrated in figure 7 is backwards. Dispersed pigment that shielded short wavelength light would depolarize the photoreceptors, and one could imagine that the different colors of chromatophore pigments would shade the photoreceptor to different degrees, causing a graded amount of activity in the cell. When the pigment is contracted within the chromatophore, the photoreceptor would hyperpolarize and turn off. I don't think that this inversion changes the predicted function of this system as a way to monitor color-change, although I haven't been able to give it enough thought to be sure. If I am incorrect, then more explication in the section with this model should be presented to clarify.

Significance:

This paper provides novel insight into the structure of the dermal photoreception and chromatophore systems, and offers a model for how, given this structure, these features interact. As stated within the manuscript, extraocular opsin expression is well established, but mechanistic/structural explanations for how opsins and dermal photoreception contribute to color-changes is sorely lacking, and so this study provides much needed information. I believe this manuscript is an important and novel contribution to this field, and is well worthy of publication.

Data and Methodology:

The data and methodology within the paper appear mostly sound. The immunohistochemistry showing SWS1 opsin labeling in a layer beneath the chromatophores is strong, as is the presence of the cells with reticulated membranes beneath the chromatophores. A small concern— I understand that immuno-gold labeling of ultrathin section is challenging, but the sparsity of the signal is quite low, a

mere 2 specks! A few different examples of the same labeling pattern from other samples, or one image with more robust labeling would greatly strengthen this piece of evidence. Both the TEM and microspectrophotometry are somewhat beyond my expertise, and so I am unable to assess their accuracy and methodology more deeply.

Suggested Improvements:

Concerning the data and model, my suggestions are written above. I don't think the authors necessarily need to do any additional experimental work before publication.

Clarity and Context:

This is the area that I think needs the most work. I found it difficult to understand where certain parts of the manuscript were going. A number of sections of the text are too vague and generalize too broadly, or don't provide enough detail that the reader requires. I've given some specific examples here, but the general suggestions should be kept in mind throughout the text. All comments below are because I like this paper and want to help it be as accessible as possible!

Line 22: cephalopod chromatophores are fundamentally different than others, and are simple organs, not cells. I don't think these examples need to be excluded, but their inclusion requires changes to ensure accuracy.

Lines 28-29: the retina and skin, and excised skin of which animals? Citations are given, but that requires the readers to stop what they're doing to look up papers.

Line 44-46: same as above— in which animals specifically has opsin levels been determined, and which animals show SWS1 activation mediating chromatophore responses to light.

Line 60-61: A modified version of this sentence should be the topic sentence for this paragraph. The point of the paragraph is what is known about the arrangements of chromatophores and dermal photoreceptors and how understanding these relationships provide insights into the function of dermal photoreception. The current topic sentence is both too vague ("regarding the localization...") and too specific ("opsins can be expressed within chromatophores..."). The reader doesn't yet know why localization matters before the descriptions of localizations begin.

Line 65: I know that hogfish change color, but a brief summary of what that looks like/what is known about color change in this species would be of great use to the general reader. Why hogfish specifically? What kind of color change happens? This is described in the figure legend somewhat, but is worth including in the main text as well.

I think there needs to be an explanation of what is known about how the dispersion of pigment happens in chromatophores. My understanding is that it can be hormonal or neural, but are there other options? Given that the change in pigment distribution is fundamental to color change as well as the functional model in the discussion, it needs to be addressed specifically.

Paragraph starting at line 74: Is this all new results/description? Has there been no previous description of the chromatophores of hogfish in prior works? If not, then it may be worth stating that this is the case. Otherwise this paragraph seems to belong in the introduction.

Paragraph starting at line 120: Be explicit about how estimating the spectral sensitivity of the hogfish SWS1 opsin is an important piece of the puzzle, why does the wavelength matter? I'm able to glean the significance after reading about the microspectrophotometry on the chromatophores, but it should be clear from the text why this is done. I'm not sure the topic sentence of this paragraph accurately represents what the paragraph is about.

Paragraph starting at line 168: Here is another place where the dispersion of the pigment is a key player in this story. Can the mechanism for how it happens be incorporated into the functional model? If not, what pieces are missing and require further investigation? I understand that this aspect isn't the main point of the paper, but again, I think it needs to be mentioned!

Paragraph starting at line 178: This paragraph mentions membrane elaboration and sensitivity, as well as a larger surface area for opsin expression. While true, the reason for devoting so much membrane to increase opsin presence in the membrane is to increase photon capture. Why might these photoreceptors require more photon capture in terms of the model? Speculation on this might be beyond the scope of the paper, but even stating that you don't know how the extra photon capture plays into the system may be worth including. Not all, and maybe not even most, extraocular photoreceptors have elaborated membranes.

Paragraph starting at line 231: This paragraph is where the proposed model comes in, and where I believe the way that the dispersion interacts with the photoreceptors is not accurately described. While it could be that this c-opsin does something different, the more simple assumption is that the transduction cascade and cell physiology is similar. If there are specific reasons to think that it's actually different, those should be stated.

Reviewer Comments

Reviewer #1 (Remarks to the Author)

R1: This manuscript investigates dermal photoreception in the hogfish, characterising for the first time in an vertebrate the optical organisation of dermal opsin. The authors demonstrate that dermal photoreceptors are positioned underneath chromatophores, suggesting that the photoreceptors may detect colour change performance. The paper is very well written and is a thorough investigation of this system. I really enjoyed reading the paper and I think it is well suited to Nature Communications (including the format - results are easily understood with the methods at the end).

A: We appreciate this positive assessment of the paper. Thank you!

R1: My comments/suggestions are all minor. Mostly, I would like a little more information about the ecology and broader function to better understand how this system may work.

A: We agree this point and provide detailed responses to each specific concern below.

1) Some more information about hogfish would be useful, such as if they use colour change for communication, camouflage or both?

A: We have now included a paragraph summarizing the current knowledge on hogfish color change ecology. This new paragraph can be found at the end of the introduction (ln 86+ in the revised manuscript), incorporating information from three new references.

This new paragraph reads: “The subject of our study, the hogfish (Perciformes: Labridae; Fig. 1), is the largest and most economically valuable wrasse of the western North Atlantic Ocean³¹. Its distinguishing features include hermaphroditic and harem reproductive strategies³², which may incorporate color change as a form of social signaling in addition to background-matching camouflage³³. Post-settlement, both males and females are capable of dynamic color change³³ (within seconds or less, Schweikert pers. observation) between at least three chromatic morphs³⁴: uniform white, uniform reddish-brown, and a mottled coloration (Fig. 1). Studies are lacking however, on the underlying physiology of hogfish color change²⁰. Here, we used approaches in immunohistochemistry, confocal and transmission electron microscopy, sequenced-based spectral sensitivity estimation, and microspectrophotometry (MSP) allowing us to investigate the physical and optical relationship between SWS1 opsin and chromatophores in hogfish skin.”

McBride, R. S. & Murphy, M. D. Current and potential yield per recruit of hogfish, *Lachnolaimus maximus*, in Florida. Pages 513–525 in Proceedings of the Gulf and Caribbean Fisheries Institute 54th Annual Meeting, Providenciales, Turks and Caicos Islands. (2003).

McBride, R. S. & Johnson, M. Sexual development and reproductive seasonality of hogfish (Labridae: *Lachnolaimus maximus*), a hermaphroditic reef fish. *J. Fish Biol.* 71, 1270-1292 (2007).

Longley, W. H. Studies upon the biological significance of animal coloration. I. The colors and color changes of West Indian reef-fishes. *J. Exp. Zool.* 23, 533-601 (1917).

2) How do animals use dermal photoreceptors? The start of the discussion really clearly lays out potential functions of dermal photoreceptors, which I think could be either moved to the introduction or stated more clearly in the introduction. Following from this, how do animals integrate the information from dermal photoreceptors? For example, if their colour change is for camouflage, is the idea that the animal integrates information from the eye about background colour with information from the dermal photoreceptors about its own colour to improve background match? Perhaps this is obvious, but it wasn't entirely clear to me.

A: We appreciate this comment and in response included new information in the introduction (lines 36+ in the revised manuscript) relating to what is presented at the start of the discussion and more clearly describes the current hypotheses about the function of dermal photoreception in color change. As part of this summary, the possible relationship of dermal photoreception to broader control by the central nervous system is stated; however, the specifics of this possibility remain unknown. This new information reads, "In live animals however, support is lacking for the direct capacity of dermal photoreception to regulate color change, leaving the function of dermal photoreception in dynamic color change unknown. One hypothesis states that dermal photosensitivity may allow for the regulation of color change independently of inputs from ocular vision¹³. The putative benefits of this strategy include reduced demands of sensory processing for color change or the possibility of light detection outside of the field-of-view or spectral sensitivity of the eyes¹³. Another hypothesis states that dermal photoreception may locally affect color change within a broader system of control that may coordinate with the central nervous system¹⁵. This possibility could allow monitoring of chromatophore color change within a feedback system, not unlike the intrinsic photosensitivity of light organs in certain mesopelagic shrimps and the bobtail squid (*Euprymna scolopes*) thought to help regulate outputs of bioluminescence^{18, 19}. For color change however, evidence is lacking to support these possibilities, leading to questions about how and why dermal photoreception and color change may be linked."

Lastly, the description of how this system might work is correct. To clarify this point to the audience, the following statement and a new supplemental figure (Fig. S3) have been added to the manuscript (ln 273 in the revised version), "Specifically, environmental cues for color change may be captured by the eyes and integrated with feedback information from dermal photoreceptors about skin color state to fine-tune color change output (Fig. S3)."

Additional Comments:

R1: Line 21: Would colour change due to changes in nanostructures also be considered dynamic colour change? Specifically, I'm

thinking of several Charidotella leaf beetles that can change colour by fluid movement between layers in the cuticle that produce structural colour. As far as I know, chromatophores aren't involved.

A: This is correct. Dynamic color change is not solely dependent on pigmentary processes, but also rearrangement of light-reflecting crystals or platelets. This is now described in the introduction (ln 26+), stating “Several major types of chromatophores exist, changing color through the intracellular reorganization of pigment granules, crystals, or reflective platelets^{8,12}. For the pigmentary chromatophores of vertebrates, pigment organelles are reversibly aggregated and dispersed within these cells by molecular motors over an extensive microtubule network^{5,8}. As a result, incident light strikes either the underlying (typically white) tissue or the exposed pigment (Fig. 1), which gives the skin its light or colored appearance, respectively⁸.”

R1: Line 33-35: This sentence could be written a little more clearly for people less familiar with the content.

A: We agree with this comment and have revised this to read (ln 50 in the revised manuscript, “Our understanding of dermal photoreception in color change is largely based on studies of gene expression (e.g., ^{20,21}), which have indicated that opsins and other phototransduction components expressed in skin have varying similarity to the phototransduction components of the retina.”

R1: Line 62: Do you have space to spell out the function a little more?

A: This line (now ln 68 in the revised manuscript) now reads “In addition to studies of gene expression, those examining the arrangement of dermal opsins relative to other components in skin provide key insights into the potential functions of dermal photoreception.”

In addition, we expanded on the potential functions of dermal photoreception by inserting a new paragraph in the introduction as described above (ln 36+ in the revised manuscript). Again here, “In live animals however, support is lacking for the direct capacity of dermal photoreception to regulate color change, leaving the function of dermal photoreception in dynamic color change unknown. One hypothesis states that dermal photosensitivity may allow for the regulation of color change independently of inputs from ocular vision¹³. The putative benefits of this strategy include reduced demands of sensory processing for color change or the possibility of light detection outside of the field-of-view or spectral sensitivity of the eyes¹³. Another hypothesis states that dermal photoreception may locally affect color change within a broader system of control that may coordinate with the central nervous system¹⁵. This possibility could allow monitoring of chromatophore color change within a feedback system, not unlike the intrinsic photosensitivity of light organs in certain mesopelagic shrimps and the bobtail squid (*Euprymna scolopes*) thought to help regulate outputs of bioluminescence^{18, 19}. For color change however, evidence is lacking to support these possibilities, leading to questions about how and why dermal photoreception and color change may be linked.”

R1: L160: This is unclear whether you are referring to the current research or previous research. It is clear in the introduction that

it has been discovered before, but could be a little clearer here since it's the opening sentence.

A: We aim to refer to the study of SWS1 in general (both previously and now), and so we have edited this sentence to read (ln 187 in the revised version) “The expression of SWS1 opsin in hogfish skin suggests dermal spectral sensitivity that coincides with the availability of short-wavelength light that predominates in their coral reef habitat, but spectrally contrasts with the long-wavelength pigmentation of their skin.”

R1: Line 213: Just out of interest, how many nanometers (roughly) are these large shifts? I am wondering if what the implication would be if one of the sites does happen to produce a large shift. I am guessing the outcome will be the same - the absorbance will still be within the region of low transmittance for the chromatophores.

A: According to SWS1 site-directed mutagenesis studies reported in Yokoyama 2008, the largest shifts (showing from amino acid substitutions at position 86) can be as high as 66 or 75nm between ultraviolet and blue light. We agree that despite this shift, the functional outcome would be similar, with the chromatophore pigment attenuating all short wavelengths of light as explained in ln251+ in the revised manuscript.

R1: Line 388: Methods are well written! Only comment is, is there a reason why this value ($\lambda_{\text{cut-off}}$) was estimated rather than calculated?

A: The reason that λ_{cut} was not calculated was because it did not formally apply to two of three chromatophore transmission spectra, with xanthophores only moderately attenuating short-wavelength light and melanophores lacking a transition between low and high transmittance. For this reason, we have removed the estimation of λ_{cut} from the paper and now describe the characteristics of the spectra instead. These methods (ln424+ in the revised manuscript) now read, “The averaged spectra were visually inspected in order to describe transitions between regions of low and high transmittance. The reported wavelengths reflect the point where the slope of these transition would intersect with the x-axis.” Correspondingly, the results have been edited to read (ln176+), “For all three chromatophore types, transmittance was positively correlated with wavelength. The mean spectra of erythrophores and xanthophores revealed sharp transitions between regions of low and high transmittance, occurring at roughly 550 nm and 488 nm, respectively (Fig. 6). By comparison, melanophores had relatively low transmittance that increased slowly and uniformly.”

R1: Figure 7: Love this, really clear!

A: Thank you!

Reviewer #2 (Remarks to the Author)

R2: This is a very interesting paper that reports the potential

cellular mechanism by which fish chromatophore pigment activity and resultant color change could be monitored by dermal photoreception. This could provide sensory feedback to fine-tune color change for a variety of behavioral functions.

A: We agree and appreciate this assessment of the paper.

R2: Comments for the authors:

Abstract: first sentence. Not sure whether referencing is allowed in Nature Comm, but the first sentence would benefit from a reference, since this forms the basis for this paper.

A: We recognize that the sweeping nature of the original statement made its basis in the literature unclear. We have revised this statement (ln 1 in the revised manuscript) to read, "Dynamic color change has evolved multiple times, with a physiological basis that has been repeatedly linked to dermal photoreception via the study of excised skin preparations." Information on these studies of skin preparations are provided in lns 34+ and 61+ of the revised manuscript.

R2: Introduction: First two sentences use the same two references. The field is much broader than represented by those references, and it would be valuable to include relevant literature on color change in fish in particular, paying attention to the overall mechanisms of color change. Pigment changes are not the only changes that occur in fish, and differentiation between pigment and structural color changes are important to point out. This is especially relevant to line 22, that suggests color change is achieved by chromatophores Reference 3 is a very old book that is of course useful but the authors may want to consider adding more recent references that deal specifically with fish chromatophores (there are other relevant papers by Sköld et al, for example, but also others).

A: We appreciate this comment and in response, have substantially revised the introduction to include information on the types of color change (morphological vs. physiological), the contributing factors (pigment vs. reflecting platelets), and regulatory mechanisms (neuronal vs. hormonal). This new information begins a ln 20+ in the revised manuscript, incorporating eight new references and reads, "Relative to morphological color change, occurring over days to months^{3,4}, dynamic or physiological color change can occur within minutes or less^{5,8}. These rate differences are based on regulation mechanisms, with the most rapid forms of color change due to neuronal rather than hormonal primary inputs of control^{5,8-10}. Animals capable of dynamic color change include cephalopods¹¹, amphibians⁷, reptiles¹, fish⁷ and others ectotherms¹, all achieving this feat using specialized skin cells called chromatophores^{1,8,12}. Several major types of chromatophores exist, changing color through the intracellular reorganization of pigment granules, crystals, or reflective platelets^{8,12}. For the pigmentary chromatophores of vertebrates, pigment organelles are reversibly aggregated and dispersed within these cells by molecular motors over an extensive microtubule network^{5,8}. As a result, incident light strikes either the underlying (typically white) tissue or the exposed pigment (Fig. 1), which gives the skin its light or colored appearance, respectively⁸."

New References:

- Sugimoto, M. Morphological color changes in fish: regulation of pigment cell density and morphology. *Microsc. Res. Tech.* 58, 496-503 (2002).
- Leclercq, E., Taylor, J. F., & Migaud, H. Morphological skin colour changes in teleosts. *Fish Fish.* 11, 159-193 (2010).
- Aspengren, S., Hedberg, D., Sköld, H. N., & Wallin, M. New insights into melanosome transport in vertebrate pigment cells. *Int. Rev. Cell. Mol. Biol.* 272, 245-302 (2009).
- Stuart-Fox, D., & Moussalli, A. Selection for social signalling drives the evolution of chameleon colour change. *Plos Biol.* 6, 22-29 (2008).
- Sköld, H. N., Aspöngren, S., Cheney, K.L., & Wallin, M. Fish chromatophores—from molecular motors to animal behavior. *Int. Rev. Cel. Mol. Bio.* 321, 171-219 (2016).
- Fujii, R. The regulation of motile activity in fish chromatophores. *Pigm. Cell Res.* 13, 300-319 (2000).
- Messenger, J. B. Cephalopod chromatophores: neurobiology and natural history. *Biol. Rev.*, 76, 473-528 (2001).
- Bagnara, J. T., Taylor, J. D., & Hadley, M. E. The dermal chromatophore unit. *J. Cell Biol.* 38, 67-79 (1968).

R2: A short section on the color change in this fish species would be good. What are color changes used for? How fast are they? What are the behavioral patterns of this fish species?

A: In response to this concern, we have now included a new paragraph summarizing the current knowledge on hogfish color change ecology. This new paragraph, incorporating three new references, reads (ln 86+): “The subject of our study, the hogfish (Perciformes: Labridae; Fig. 1), is the largest and most economically valuable wrasse of the western North Atlantic Ocean³¹. Its distinguishing features include hermaphroditic and harem reproductive strategies³², which may incorporate color change as a form of social signaling in addition to background-matching camouflage³³. Post-settlement, both males and females are capable of dynamic color change³³ (within seconds or less, Schweikert pers. observation) between at least three chromatic morphs³⁴: uniform white, uniform reddish-brown, and a mottled coloration (Fig. 1). Studies are lacking however, on the underlying physiology of hogfish color change²⁰. Here, we used approaches in immunohistochemistry, confocal and transmission electron microscopy, sequenced-based spectral sensitivity estimation, and microspectrophotometry (MSP) allowing us to investigate the physical and optical relationship between SWS1 opsin and chromatophores in hogfish skin.”

R2: Results: How fast are the color changes? Are reflector cells involved? Fig. 1b shows some blue iridescent looking structures that seem to suggest that structural coloration is involved? I would like to see the spectral data (Fig. 6) in the first paragraph of the results, incorporated into Fig. 1, possibly.

A: The rate of hogfish color change has never been published. From personal observation, it can occur within one second or less, which is now stated in the manuscript in ln 90 of the revised manuscript. The involvement of reflector cells is unknown, but in response to this comment, we have inserted the following text (ln 105+ in the revised manuscript: “Observation of white

reflectivity and blue iridescence in the aggregated pigment preparations (Fig. 1b) suggests the presence of leucophores or iridophores in hogfish skin; however, the presence of these chromatophore types have yet to be observed in our analyses by transmission electron micrography.”

In addition, we considered the request to incorporate Figure 6 into Figure 1. We understand the rationale for the suggestion (further provided below); however, we have decided to maintain the order of the results for clarity of the study narrative. That is, the only reason we were motivated to measure the chromatophore transmission spectra was because the SWS1 receptors were discovered underneath the chromatophore pigment. In aiming to determine the consequences of having receptors beneath pigment, it is critical that the SWS1 immunohistochemistry be presented prior to the microspectrophotometry results (Fig. 6). To further clarify our rationale to the audience, we have edited the topical sentence of the microspectrophotometry results section to read (Ins 168+ in the revised manuscript), “The alignment of chromatophores over putative photoreceptors (Fig. 3) indicates that ambient light must first pass through chromatophores before striking SWS1 opsin in hogfish skin, and thus, we were interested in determining the effects of chromatophore pigment on light transmission using microspectrophotometry (MSP).”

R2: The immunogold labeling images are not terribly convincing - perhaps you could comment on this in the paper, if you do not have more convincing images? Maybe show several images that prove labeling?

A: In response to this comment, we added new images of the immunogold labeling to Figure 4 and revised the methods to explain how our selected approach resulted in relatively sparse labeling by the secondary antibody as expected. Explained by Cornford and colleagues (2003), a tradeoff exists between colloidal gold particle size and the number of antigenic sites, with larger sizes resulting in sparser labeling (see Cornford et al. 2003 Fig. 2 below). These large particle sizes (as used here; 25-nm gold) however, permit the low-magnification imaging required to visualize large target cells (i.e., chromatophores and underlying cells). Thus, the density of secondary antibody in our immunogold micrographs is in line with expectations. The new language in the methods (ln 376+) reads, “As expected, use of these relatively large gold particles permitted the low-magnification imaging required to visualize the target cells but resulted in sparse labeling by the secondary antibody as explained by Cornford and colleagues (2003)50.”

Cornford, E. M., Hyman, S., & Cornford, M. E. (2003). Immunogold detection of microvascular proteins in the compromised blood-brain barrier. In *The Blood-Brain Barrier* (pp. 161–175). Humana Press.

Fig. 2. Comparison of GFAP epitope identification and gold particle size in pericapillary fibrils from a human brain resection. Sequential sections from the same tissue block were mounted on separate grids, to compare epitope densities with either 20-nm gold particles (*left panel*), 10-nm gold (*center panel*), or 5-nm gold particles (*right panel*). In both the left and right panels, a mouse monoclonal antiserum to human GFAP was employed, together with 5-nm or 20-nm gold-labeled goat antimouse sera. In the center panel, a rabbit polyclonal antibody to bovine GFAP was used in conjunction with 10-nm gold-labeled goat antirabbit serum. Note that when smaller sized gold particles are used, relatively more antigenic sites are apparent.

R2: Line 108 - the cells were found beneath each adjacent chromatophore cell: are these cells absent from places that are lacking chromatophores? Such places may be hard to find in this fish, which I'm not completely familiar with, but perhaps an area between the fin spines, ventral belly, underneath throat?

A: This is a clever idea that could provide a negative control for the coupling of chromatophores and dermal photoreceptors. Through visual inspection of hogfish however, we cannot find an area of skin devoid of chromatophores including between the fin spines, on the ventral belly or throat. We might opportunistically find such an area in future cross-sectional assessments of skin, but for now, we revised the manuscript to say (ln 136+) "The two [cell types] were always coupled, with no instances of membrane-filled cell lacking an overlying chromatophore."

R2: MSP data: I would move this up to the beginning of the results section, and incorporate the Fig. 6 into Fig. 1. Could you add how many pigment granules were measured in the transmission graphs? Presumably, n=60 refers to the number of chromatophores? It may be worth adding images to the Figure that show the pigment coverage that was measured?

A: We appreciate this comment from the reviewer. Our decision to decline the suggested edit to Figure 1 is described above, but again here: we have decided to maintain the order of the results for clarity of the study narrative. That is, the only reason we were motivated to measure the chromatophore transmission spectra was because the SWS1 receptors were discovered underneath the chromatophore pigment. In aiming to determine the consequences of having receptors beneath pigment, it is critical that the SWS1 immunohistochemistry be presented prior

to the microspectrophotometry results (Fig. 6). To further clarify our rationale to the audience, we have edited the topical sentence of the microspectrophotometry results section to read (lns 168+ in the revised manuscript), “The alignment of chromatophores over putative photoreceptors (Fig. 3) indicates that ambient light must first pass through chromatophores before striking SWS1 opsin in hogfish skin, and thus, we were interested in determining the effects of chromatophore pigment on light transmission using microspectrophotometry (MSP).”

Unfortunately, we are unable to report the number of pigment granules measured. The pigment granules, at least for melanophores, are packaged within dense melanosomes and are unable to be reliably quantified for individual cells using the microscopy techniques reported here. The reviewer is correct that “n=60” refers to the number of each chromatophore type measured. To clarify this, ln 172 now reads, “(n = 60 cells each for melanophores, erythrophores, and xanthophores).” Also, we are unable to produce an image that shows the pigment coverage that was measured for individual cells by MSP, but we worked to more thoroughly describe this coverage with the following revision (lns 416+ in the revised manuscript), “For each measurement light, from a condensing objective was focused through a single chromatophore cell (to the margins of dispersed pigment), then onto a 1-mm diameter fiber (Ocean Optics Inc., Dunedin, FL) connected to a USB2000 spectrometer (Ocean Optics Inc., Dunedin, FL).”

R2: Discussion: Lines 161-163 talks about the short-wave sensitivity of the photoreception and attempts to bring this into ecological context. A short-wave sensitivity at 415 nm is, in my opinion, of very limited use. All of the chromatophores are long-wavelength pigments and it seems that long-wave photosensitivity would make more sense. Perhaps you could expand on this? Unless, of course, the short-wave sensitivity is involved with other structures, e.g., short-wave structure colors (see Fig. 1b).

A: To clarify, the short wavelength sensitivity of the receptor is a critical characteristic for it to be sensitive to changes by a long-wavelength transmitting (red) chromatophore. If the receptor was long-wavelength sensitive, as you suggest here, there would be little change in receptor irradiation either when pigment was aggregated (and red light could pass) or when the pigment was dispersed (still allowing red light to pass). In other words, if red sensitive, the receptor would have less information that color change was happening. Thus, fundamentally, it is the sensitivity of the receptor to the light that is best attenuated by the pigment (i.e., blue) that allows it to experience the effects of color change. This is currently explained in lns 195+ and 261+ of the revised manuscript. We considered including the antithetical explanation provided here, but were afraid it would create confusion.

R2: Also, I found the low transmission of the chromatophore pigments very intriguing; approx. 50% for yellow xanthophores, 5-15% for melanophores and erythrophores. Under what conditions (if so) do they disperse separately? It seems that they disperse together (Fig. 1b), or are these fish spp. able to differentially disperse different chromatophore classes? One possible scenario could be that the differential activation of the photosensitive cells provided by the differential transmission from the chromatophores could be the trigger for fine tuning of color change? See also your discussion points in lines 202-208.

Lines 222-225, see my last comment. How could this help with regulating color change? It's worth expanding here, in my opinion. Lines 236-237. Fine-tune color change performance. This sentence (while it can stay where it is), needs to be explained somewhere in the introduction. At this point, the reader does not fully understand why this fish species changes color, and why the color may need to be fine tuned. While some color changes in fish are fast, they usually take hours or days to complete. It would be good to expand on this in the context of skin photoreception.

A: It is currently unknown if hogfish are capable of differentially dispersing different chromatophore classes or how this might contribute to fine tuning of color change, though they appear to differentially disperse chromatophores over the body resulting in the mottled coloration indicated in Figure 1. As for the reviewer's comment about original manuscript lns 202-208 and 222-225, we repeat the following explanation: To clarify, the short wavelength sensitivity of the receptor is a critical characteristic for it to be sensitive to changes by a long-wavelength-transmitting (red) chromatophore. If the receptor was long-wavelength sensitive, as you suggest here, there would be little change in receptor irradiation either when pigment was aggregated (and red light could pass) or when the pigment was dispersed (still allowing red light to pass). In other words, if red-sensitive, the receptor would have less information that color change was happening.

We appreciate the final request for clarity about how and why color change must be fine tuned. To address this, we have added two new sections of information and a new supplemental figure (Fig. S3). In the introduction, lns 36+ now reads "In live animals however, support is lacking for the direct capacity of dermal photoreception to regulate color change, leaving the function of dermal photoreception in dynamic color change unknown. One hypothesis states that dermal photosensitivity may allow for the regulation of color change independently of inputs from ocular vision¹³. The putative benefits of this strategy include reduced demands of sensory processing for color change or the possibility of light detection outside of the field-of-view or spectral sensitivity of the eyes¹³. Another hypothesis states that dermal photoreception may locally affect color change within a broader system of control that may coordinate with the central nervous system¹⁵. This possibility could allow monitoring of chromatophore color change within a feedback system, not unlike the intrinsic photosensitivity of light organs in certain mesopelagic shrimps and the bobtail squid (*Euprymna scolopes*) thought to help regulate outputs of bioluminescence^{18, 19}. For color change however, evidence is lacking to support these possibilities, leading to questions about how and why dermal photoreception and color change may be linked."

In the discussion, ln 173+ now reads, "Specifically, environmental cues for color change may be captured by the eyes and integrated with feedback information from dermal photoreceptors about skin color state to fine-tune color change output (Fig. S3). Such closed-loop feedback systems are common in physiology and behavior⁴⁴ and may be required by color change as they are by other outputs where fitness is coupled to the precision of performance^{45,46}."

Reviewer #3 (Remarks to the Author)

Key results:

R3: Schweikert et al. address the question of how a dermal photoreception system may interact with chromatophores to help explain how hogfish color-change is controlled. They found SWS1 opsin expression in a layer of cells that lie beneath the chromatophores, rather than within the chromatophores themselves using a combination of immunohistochemistry and TEM. The authors estimated that the SWS1 opsin in hogfish skin has a peak sensitivity at short wavelength, 415nm light, as the sequence is highly similar to SWS1 opsins from the retinas of other fish species. They found that all 3 types of chromatophores in hogfish skin interfere with the amount of short wavelength light that would otherwise reach the photoreceptors when the chromatophore pigment is dispersed, although the extent of this interference differs between chromatophores of different colors.

A: We appreciate and agree with this interpretation of our paper.

Validity:

R3: For the most part, the interpretation of the data and conclusions seem sound. That said, given my understanding of opsin phototransduction cascades and their physiological effects within photoreceptor cells, I believe the model presented in the paper for how the chromatophores interact with photoreceptors is not accurate. Typically, when ciliary opsins are exposed to light, they cause the cell to hyperpolarize, effectively turning it "off", and the cells depolarize (turn "on") in the dark. Based on this understanding, the model presented in the discussion and illustrated in figure 7 is backwards. Dispersed pigment that shielded short wavelength light would depolarize the photoreceptors, and one could imagine that the different colors of chromatophore pigments would shade the photoreceptor to different degrees, causing a graded amount of activity in the cell. When the pigment is contracted within the chromatophore, the photoreceptor would hyperpolarize and turn off. I don't think that this inversion changes the predicted function of this system as a way to monitor color-change, although I haven't been able to give it enough thought to be sure. If I am incorrect, then more explication in the section with this model should be presented to clarify.

A: We appreciate this comment and the consideration of our findings in the context of retinal photoreceptor physiology. We have worked to address these concerns to the best of our ability both in the response below and in the manuscript. Ultimately, we have decided to take the more conservative approach of not aligning to a given possibility regarding cell activation (versus assuming similar physiology between retinal and dermal photoreceptors) for reasons that include the following. First, the activation of ciliary opsins in retinal photoreceptors activates a cGMP signaling cascade that causes cell hyperpolarization, there is no guarantee that ciliary opsin activation in dermal photoreceptors (thought to occur via a cAMP dependent cascade; stated in ln 54+ in the revised manuscript) would result in the same outcome. Second, regardless of whether the dermal photoreceptors hyperpolarize or depolarize in response to light, the resultant feedback

on chromatophores be would dependent on the type of synaptic connections and presence of excitatory or inhibitory signaling molecules, which are also currently unknown. We now explicitly list these knowledge gaps in Ln 267+ (in the revised manuscript), reading “One missing piece however, is determining how dermal photoreceptors communicate with chromatophores to exert feedback control on skin color change. For example, the activation characteristics of the SWS1 receptors are unknown, along with the synapses and signaling molecules that may connect the two cell types.”

In regard to Figure 7, because irradiation would result in the activation of the SWS1 opsin (if not the entire cell), we would like to keep the SWS1 “ON” for the aggregated state. To clarify however, we’ve ensured that we only discuss opsin activation (and not the entire cell), including in Figure 7 legend now reading, “Dispersed chromatophore pigment suppresses short-wavelength irradiation of SWS1 receptors (left), whereas aggregated pigment permits short-wavelength irradiation (and therefore, putative opsin activation) of SWS1 receptors (right)”. Also, to more accurately describe retinal photoreceptor physiology, we edited Ln 281+ to read, “For example, rod and cone characteristics, such as cell stimulation over graded membrane potentials that scale with exposure to light, would be particularly relevant to this system where the intensity of incident light upon dermal photoreceptors is dependent on the degree of pigment aggregation and chromatophore pigment type.”

As you state, the inversion of cell activation with the aggregated chromatophore state would not have a bearing on the predicted feedback function and again, would be dependent on physiological characteristics of this system yet to be described.

Significance:

R3: This paper provides novel insight into the structure of the dermal photoreception and chromatophore systems, and offers a model for how, given this structure, these features interact. As stated within the manuscript, extraocular opsin expression is well established, but mechanistic/structural explanations for how opsins and dermal photoreception contribute to color-changes is sorely lacking, and so this study provides much needed information. I believe this manuscript is an important and novel contribution to this field, and is well worthy of publication.

A: We agree with and appreciate this positive assessment of our paper.

Data and Methodology:

R3: The data and methodology within the paper appear mostly sound. The immunohistochemistry showing SWS1 opsin labeling in a layer beneath the chromatophores is strong, as is the presence of the cells with reticulated membranes beneath the chromatophores. A small concern— I understand that immuno-gold labeling of ultrathin section is challenging, but the sparsity of the signal is quite low, a mere 2 specks! A few different examples of the same labeling pattern from other samples, or one image with more robust labeling would greatly strengthen this piece of evidence. Both the TEM and microspectrophotometry are somewhat beyond my expertise, and so I am unable to assess their accuracy and methodology more deeply.

A: In response to this comment, we added new images of the immunogold labeling to Figure 4 and revised the methods to explain how our selected approach resulted in relatively sparse labeling by the secondary antibody as expected. Explained by Cornford and colleagues (2003), a tradeoff exists between colloidal gold particle size and the number of antigenic sites, with larger sizes resulting in sparser labeling (see Cornford et al. 2003 Fig. 2 below). These large particle sizes (as used here; 25-nm gold) however, permit the low-magnification imaging required to visualize large target cells (i.e., chromatophores and underlying cells). Thus, the density of secondary antibody in our immunogold micrographs is in line with expectations. The new language in the methods (ln 376+), “As expected, use of these relatively large gold particles permitted the low-magnification imaging required to visualize the target cells but resulted in sparse labeling by the secondary antibody as explained by Cornford and colleagues (2003)50.”

Cornford, E. M., Hyman, S., & Cornford, M. E. (2003). Immunogold detection of microvascular proteins in the compromised blood-brain barrier. In *The Blood-Brain Barrier* (pp. 161–175). Humana Press.

Fig. 2. Comparison of GFAP epitope identification and gold particle size in pericapillary fibrils from a human brain resection. Sequential sections from the same tissue block were mounted on separate grids, to compare epitope densities with either 20-nm gold particles (*left panel*), 10-nm gold (*center panel*), or 5-nm gold particles (*right panel*). In both the left and right panels, a mouse monoclonal antiserum to human GFAP was employed, together with 5-nm or 20-nm gold-labeled goat antimouse sera. In the center panel, a rabbit polyclonal antibody to bovine GFAP was used in conjunction with 10-nm gold-labeled goat antirabbit serum. Note that when smaller sized gold particles are used, relatively more antigenic sites are apparent.

Suggested Improvements:

R3: Concerning the data and model, my suggestions are written above. I don't think the authors necessarily need to do any additional experimental work before publication.

A: We appreciate this assessment of our paper.

Clarity and Context:

R3: This is the area that I think needs the most work. I found it difficult to understand where certain parts of the manuscript were going. A number of sections of the text are too vague and

generalize too broadly, or don't provide enough detail that the reader requires. I've given some specific examples here, but the general suggestions should be kept in mind throughout the text. All comments below are because I like this paper and want to help it be as accessible as possible!

A: We appreciate the effort expended to help improve the following aspects of the paper.

R3: Line 22: cephalopod chromatophores are fundamentally different than others, and are simple organs, not cells. I don't think these examples need to be excluded, but their inclusion requires changes to ensure accuracy.

A: This is now addressed. The original manuscript attempted to provide a generalized description that applied to all chromatophores, which we now see is confusing. In response, we clarify our description of vertebrate chromatophores in ln 27+ (in the revised manuscript), which reads: "For the pigmentary chromatophores of vertebrates, pigment organelles are reversibly aggregated and dispersed within these cells by molecular motors over an extensive microtubule network^{5,8}. As a result, incident light strikes either the underlying (typically white) tissue or the exposed pigment (Fig. 1), which gives the skin its light or colored appearance, respectively⁸."

Then, we clarify our description of cephalopod chromatophores in lins 88 (revised manuscript), now reading: "In the inshore squid (*Doryteuthis pealeii*), rhodopsin has been localized to several cell types comprising chromatophore organs: the pigment cells, radial muscle fibers, and sheath cells, which may individually or synergistically respond to incident light¹⁵."

R3: Lines 28-29: the retina and skin, and excised skin of which animals? Citations are given, but that requires the readers to stop what they're doing to look up papers.

A: This is now addressed, now reading (lins 34+ revised manuscript): "Evidence of dermal photoreception, such as for cephalopods and fish, includes phototransduction proteins (e.g., opsins) of the retina co-occurring in the skin¹⁴⁻¹⁶ and incident light on excised skin patches inducing a color-change response^{16,17}."

R3: Line 44-46: same as above— in which animals specifically has opsin levels been determined, and which animals show SWS1 activation mediating chromatophore responses to light.

A: This is now addressed. These lines now read (lins 61+) in the revised manuscript, "These studies have shown that, relative to other opsins, SWS1 opsin can have the highest expression levels in skin (in hogfish and others)^{16,20} and that SWS1 activation, at least within *in vitro* skin preparations of the Nile tilapia, can directly mediate chromatophore responses to light²⁷."

R3: Line 60-61: A modified version of this sentence should be the topic sentence for this paragraph. The point of the paragraph is what is known about the arrangements of chromatophores and dermal photoreceptors and how understanding these relationships provide insights into the function of dermal photoreception. The current

topic sentence is both too vague ("regarding the localization...") and too specific ("opsins can be expressed within chromatophores..."). The reader doesn't yet know why localization matters before the descriptions of localizations begin.

A: This is now addressed. The topical sentence has been modified as proposed (ln 68 in the revised manuscript), which now reads, "In addition to studies of gene expression, those examining the arrangement of dermal opsins relative to other components in skin provide key insights into the potential functions of dermal photoreception. "

R3: Line 65: I know that hogfish change color, but a brief summary of what that looks like/what is known about color change in this species would be of great use to the general reader. Why hogfish specifically? What kind of color change happens? This is described in the figure legend somewhat, but is worth including in the main text as well.

A: This is now addressed by the inclusion of a new paragraph in the introduction (ln 86+) that reads, "The subject of our study, the hogfish (Perciformes: Labridae; Fig. 1), is the largest and most economically valuable wrasse of the western North Atlantic Ocean³¹. Its distinguishing features include hermaphroditic and harem reproductive strategies³², which may incorporate color change as a form of social signaling in addition to background-matching camouflage³³. Post-settlement, both males and females are capable of dynamic color change³³ (within seconds or less, Schweikert pers. observation) between at least three chromatic morphs³⁴: uniform white, uniform reddish-brown, and a mottled coloration (Fig. 1). Studies are lacking however, on the underlying physiology of hogfish color change²⁰. Here, we used approaches in immunohistochemistry, confocal and transmission electron microscopy, sequenced-based spectral sensitivity estimation, and microspectrophotometry (MSP) allowing us to investigate the physical and optical relationship between SWS1 opsin and chromatophores in hogfish skin."

R3: I think there needs to be an explanation of what is known about how the dispersion of pigment happens in chromatophores. My understanding is that it can be hormonal or neural, but are there other options? Given that the change in pigment distribution is fundamental to color change as well as the functional model in the discussion, it needs to be addressed specifically.

A: This is now addressed by the inclusion of a new information in the introduction (lns 20+) that reads, "Relative to morphological color change, occurring over days to months^{3,4}, dynamic or physiological color change can occur within minutes or less^{5,8}. These rate differences are based on regulation mechanisms, with the most rapid forms of color change due to neuronal rather than hormonal primary inputs of control^{5,8-10}. Animals capable of dynamic color change include cephalopods¹¹, amphibians⁷, reptiles¹, fish⁷ and others ectotherms¹, all achieving this feat using specialized skin cells called chromatophores^{1,8,12}. Several major types of chromatophores exist, changing color through the intracellular reorganization of pigment granules, crystals, or reflective platelets^{8,12}. For the pigmentary chromatophores of vertebrates, pigment organelles are reversibly aggregated and dispersed within these cells by molecular motors over an extensive

microtubule network^{5,8}. As a result, incident light strikes either the underlying (typically white) tissue or the exposed pigment (Fig. 1), which gives the skin its light or colored appearance, respectively⁸.”

R3: Paragraph starting at line 74: Is this all new results/description? Has there been no previous description of the chromatophores of hogfish in prior works? If not, then it may be worth stating that this is the case. Otherwise this paragraph seems to belong in the introduction.

A: This is now addressed by the inclusion of a new statement in the introduction (ln 92 revised manuscript), stating “Studies are lacking however, on the underlying physiology of hogfish color change,” and by ln 100+ at the start of the results section, “Three types of chromatophores with differing pigments were identified by light microscopy of *en face* preparations of hogfish skin: black melanophores, red erythrophores, and yellow xanthophores (Fig. 1).”

R3: Paragraph starting at line 120: Be explicit about how estimating the spectral sensitivity of the hogfish SWS1 opsin is an important piece of the puzzle, why does the wavelength matter? I’m able to glean the significance after reading about the microspectrophotometry on the chromatophores, but it should be clear from the text why this is done. I’m not sure the topic sentence of this paragraph accurately represents what the paragraph is about.

A: This is now addressed by the revision of the topical sentence (ln 150 in the revised manuscript) reading, “To begin exploring the optical effects of chromatophores overlying the SWS1 receptors, we used a sequence alignment technique to estimate spectral sensitivity of the hogfish SWS1 opsin.”

R3: Paragraph starting at line 168: Here is another place where the dispersion of the pigment is a key player in this story. Can the mechanism for how it happens be incorporated into the functional model? If not, what pieces are missing and require further investigation? I understand that this aspect isn’t the main point of the paper, but again, I think it needs to be mentioned!

A: This is now addressed by the inclusion of a new information in the introduction (lns 26+) that reads, “Several major types of chromatophores exist, changing color through the intracellular reorganization of pigment granules, crystals, or reflective platelets^{8,12}. For the pigmentary chromatophores of vertebrates, pigment organelles are reversibly aggregated and dispersed within the cell by molecular motors over an extensive microtubule network^{5,8}”

How this information on chromatophore pigment motility relates to the proposed model depends on the physiological details of how the SWS1 receptors might signal to chromatophores (described as unknown in lns 267+ revised manuscript), which is difficult for us to further comment on here.

R3: Paragraph starting at line 178: This paragraph mentions

membrane elaboration and sensitivity, as well as a larger surface area for opsin expression. While true, the reason for devoting so much membrane to increase opsin presence in the membrane is to increase photon capture. Why might these photoreceptors require more photon capture in terms of the model? Speculation on this might be beyond the scope of the paper, but even stating that you don't know how the extra photon capture plays into the system may be worth including. Not all, and maybe not even most, extraocular photoreceptors have elaborated membranes.

A: This is now addressed in line 216+ of the revised manuscript which reads “The reason for this high surface area (and putative enhancement in sensitivity) is unknown but may relate to maintaining dermal photosensitivity under dim light levels that hogfish may experience when reaching oceanic depths of 20 to 45 m or more⁴².”

R3: Paragraph starting at line 231: This paragraph is where the proposed model comes in, and where I believe the way that the dispersion interacts with the photoreceptors is not accurately described. While it could be that this c-opsin does something different, the more simple assumption is that the transduction cascade and cell physiology is similar. If there are specific reasons to think that it's actually different, those should be stated.

A: Again, we appreciate this comment and our need to make several adjustments. Our response is provided again: We have worked to address these concerns to the best of our ability both in the response below and in the manuscript. Ultimately, we have decided to take the more conservative approach of not aligning to a given possibility regarding cell activation (versus assuming similar physiology between retinal and dermal photoreceptors) for reasons that include the following. First, the activation of ciliary opsins in retinal photoreceptors activates a cGMP signaling cascade that causes cell hyperpolarization, there is no guarantee that ciliary opsin activation in dermal photoreceptors (thought to occur via a cAMP dependent cascade; stated in ln 54+ in the revised manuscript) would result in the same outcome. Second, regardless of whether the dermal photoreceptors hyperpolarize or depolarize in response to light, the resultant feedback on chromatophores would be dependent on the type of synaptic connections and presence of excitatory or inhibitory signaling molecules, which are also currently unknown. We now explicitly list these knowledge gaps in ln 267+ (in the revised manuscript), reading “One missing piece however, is determining how dermal photoreceptors communicate with chromatophores to exert feedback control on skin color change. For example, the activation characteristics of the SWS1 receptors are unknown, along with the synapses and signaling molecules that may connect the two cell types.”

In regard to Figure 7, because irradiation would result in the activation of the SWS1 opsin (if not the entire cell), we would like to keep the SWS1 “ON” for the aggregated state. To clarify however, we've ensured that we only discuss opsin activation (and not the entire cell), including in Figure 7 legend now reading, “Dispersed chromatophore pigment suppresses short-wavelength irradiation of SWS1 receptors (left), whereas aggregated pigment permits short-wavelength irradiation (and therefore, putative opsin activation) of SWS1 receptors (right)”.

Also, to more accurately describe retinal photoreceptor physiology, we edited ln 281+ to read, “For example, rod and cone characteristics, such as cell stimulation over graded membrane potentials that scale with exposure to light, would be particularly relevant to this system where the intensity of incident light upon dermal photoreceptors is dependent on the degree of pigment aggregation and chromatophore pigment type.”

As you state, the inversion of cell activation with the aggregated chromatophore state would not have a bearing on the predicted feedback function and again, would be dependent on physiological characteristics of this system yet to be described.

Reviewers' Comments:

Reviewer #1:

Remarks to the Author:

Thank you to the authors for addressing mine and the other reviewers' comments so thoughtfully. I appreciate the explanations and changes to the manuscript and I do not have any other remarks. I look forward to seeing it in print.

Reviewer #2:

Remarks to the Author:

I have no further comments. The authors have addressed all my queries elegantly in the revised manuscript. This paper is a very nice piece of work and I am looking forward to seeing it published.

Reviewer #3:

Remarks to the Author:

My concerns and comments, as well as those of the other reviewers, have been well addressed by the additional changes made by the authors. I support the acceptance and publication of this manuscript with the changes made.